# Molecular mechanism of exchange coupling in CLC chloride/proton antiporters

Deniz Aydin [1,2,3,4,12], Chih-Ta Chien [5,6,11,12], Jürgen Kreiter [1,12], Amy R. Nava[1,12], Jasmina M. Portasikova [7,8], Lukas Fojtik[7,8], Briana L. Sobecks[1,2,3,4,9], Catalina Mosquera[1], Petr Man [8], Ron O. Dror [1,2,3,4] ✉, Wah Chiu [5,6,10] ✉ & Merritt Maduke [1] ✉

The ubiquitous CLC membrane transporters are unique in their ability to exchange anions for cations. Despite extensive study, there is no mechanistic model that fully explains their 2:1 $Cl^-/H^+$ stoichiometric exchange mechanism. Here, we provide such a model. Using differential hydrogen-deuterium exchange mass spectrometry, cryo-EM structure determination, and molecular dynamics simulations, we uncovered conformational dynamics in CLC-ec1, a bacterial CLC homolog that has served as a paradigm for this family of transporters. Simulations based on a cryo-EM structure at pH 3 revealed critical steps in the transport mechanism, including release of $Cl^-$ ions to the extracellular side, opening of the inner gate, and water wires that facilitate $H^+$ transport. Surprisingly, these water wires occurred independently of $Cl^-$ binding, prompting us to reassess the relationship between $Cl^-$ binding and $Cl^-/H^+$ coupling. Using isothermal titration calorimetry and quantitative flux assays on mutants with reduced $Cl^-$ binding affinity, we conclude that, while $Cl^-$ binding is necessary for coupling, even weak binding can support $Cl^-/H^+$ coupling. By integrating our findings with existing literature, we establish a complete and efficient CLC 2:1 $Cl^-/H^+$ exchange mechanism.

The CLC chloride channels are a biophysically fascinating gene family consisting of both passive chloride ($Cl^-$) channels and active transporters that exchange chloride for protons ($H^+$)[1,2]. CLC transporters are ubiquitously expressed in intracellular membranes, and their dysfunction disrupts ion homeostasis and causes Dent's disease, osteopetrosis, and neurodegenerative disorders[3]. CLC transporters are also found widely in plants and microorganisms[3]. In bacteria, CLC transporters support pathogenesis by facilitating extreme acid tolerance[4,5].

CLC transporters catalyze $Cl^-/H^+$ exchange transport with a stoichiometry of 2 $Cl^-$ ions per $H^+$. This 2:1 stoichiometry is observed in all CLC transporter homologs under all experimental conditions measured, including $Cl^-$ and $H^+$ concentrations that vary by orders of magnitude. CLC-ec1 is an *E. coli* homolog that was initially examined due to its functional similarity to mammalian homologs, together with its superior suitability for structural analysis[6,7]. CLC-ec1 was the first CLC demonstrated to be an active transporter instead of a channel[1],

[1]Department of Molecular and Cellular Physiology, Stanford University, Stanford, CA, USA. [2]Department of Computer Science, Stanford University, Stanford, CA, USA. [3]Department of Structural Biology, Stanford University, Stanford, CA, USA. [4]Institute for Computational and Mathematical Engineering, Stanford University, Stanford, CA, USA. [5]Department of Bioengineering, Stanford University, Stanford, CA, USA. [6]Department of Microbiology and Immunology, Stanford University, Stanford, CA, USA. [7]Department of Biochemistry, Faculty of Science, Charles University, Prague, Czech Republic. [8]Institute of Microbiology of the Czech Academy of Sciences, Division BioCeV, Prumyslova 595, Vestec, Czech Republic. [9]Department of Chemical Engineering, Stanford University, Stanford, CA, USA. [10]Division of CryoEM and Bioimaging, SSRL, SLAC National Accelerator Laboratory, Stanford University, Menlo Park, CA, USA. [11]Present address: National Institute of Diabetes and Digestive and Kidney Diseases (NIDDK), National Institutes of Health, Bethesda, MD, USA. [12]These authors contributed equally: Deniz Aydin, Chih-Ta Chien, Jürgen Kreiter, Amy R. Nava. ✉e-mail: ron.dror@stanford.edu; wahc@stanford.edu; maduke@stanford.edu

and has proven valuable for guiding studies of mammalian CLCs, both transporters and channels[3,8]. CLC-ec1 remains a useful paradigm, as it is currently the only transporter homolog for which multiple conformations have been determined at high resolution[7,9–11] and the only homolog whose ion-binding properties have been quantified by isothermal titration calorimetry[12,13]. In addition, the availability of a functional cysteine-less variant makes it amenable to spectroscopic approaches[14–16].

Studies of CLC-ec1's structure, function, and dynamics have provided essential clues to the CLC Cl⁻/H⁺ transport mechanism. Structurally, CLCs are homodimers in which each subunit can act independently[3]. Each subunit contains permeation pathways for Cl⁻ and H⁺, which are shared along the extracellular section and then diverge toward the intracellular side (Fig. 1a). The Cl⁻ pathway is gated by a conserved glutamate ($E_{gate}$ – E148 in CLC-ec1), which also functions as a H⁺-transfer site (Fig. 1a). This pathway is defined by two anion-binding sites, $S_{ext}$ and $S_{cen}$, in the middle of each subunit. These sites can be occupied by the negatively charged $E_{gate}$ side chain ("middle" and "down" conformations in Fig. 1b) or by Cl⁻ ions. The H⁺ permeation pathway diverges from the Cl⁻ pathway toward the intracellular side, as illustrated in Fig. 1a. This positioning of the H⁺ pathway is supported by several lines of evidence. First, mutations along the intracellular segment of the pathway, particularly at $Glu_{in}$ (E203), severely repress H⁺ transport[17,18]. Second, $E_{gate}$ can occupy this proposed H⁺ pathway in its "out" conformation (see Fig. 1b). Third, molecular dynamics (MD) simulations on CLC-ec1 detect water wires (chains of hydrogen-bonded water molecules) that provide a pathway for H⁺ conduction in this region[10,11,18–25]. Within this framework, the key to achieving 2:1 Cl⁻/H⁺ coupling must lie in the coupling of the $E_{gate}$ rotameric conformational changes to Cl⁻ and H⁺ movement in opposite directions through the pathways.

Although some important aspects of this coupling are understood —such as the correlation between coupling and Cl⁻ binding to $S_{cen}$[12,18,23,26,27] and the synergistic binding of Cl⁻ and H⁺[12,13]—a satisfactory model to fully explain the 2:1 exchange mechanism has remained elusive. Existing models explain some of CLC-ec1's known properties but fall short of capturing the full picture. For instance, proposed models tend to be coherent in only one direction of the transport cycle[10,24,28], whereas CLC-ec1 facilitates transmembrane Cl⁻/H⁺ exchange with similar efficiency in both directions[29]. Furthermore, all available structures show the inner gate of the Cl⁻ pathway as closed, leaving the mechanism of Cl⁻ movement to and from the intracellular side unresolved.

A major challenge in the study of CLC antiporters is the apparently subtle nature of the protein conformational changes, which is in stark contrast to the large conformational changes that are seen in other secondary active transporters. For instance, the crystal structure of WT CLC-ec1 shows $E_{gate}$ in the "middle" conformation (Fig. 1b). When $E_{gate}$ is mutated to glutamine (to simulate the protonation that occurs as part of the transport cycle), the resulting structure is nearly indistinguishable from that of WT, differing only in the rotation of the side chain (from "middle" to "up"), and with the extracellular pathway still narrower than a Cl⁻ ion. This observation raised an important question: are conformational changes in CLC transporters inherently subtle compared to those in other transporters, or do significant conformational shifts remain undiscovered? This question drove us, and others, to explore these changes outside the constraints of crystallization[16,30–34]. To do so, we utilized low-resolution spectroscopic techniques to analyze conformational dynamics in $E_{gate}$ mutant proteins, as well as in wild-type proteins under conditions of lowered pH to facilitate the protonation of $E_{gate}$. Through these studies, we identified a mutant (QQQ) that was stabilized in the protonated state, allowing us to determine its high-resolution structure and reveal molecular details of conformational change, including widening of the extracellular pathway and the positioning of $E_{gate}$ in the "out" conformation[10]. Subsequently, structures of WT CLC-ec1 at pH 4.5 recapitulated many of the conformational features observed in the QQQ structure as well as conformational changes in the N-terminal region and at the dimer interface[11]. However, none of these structures illuminates the mechanism of inner-gate opening. This void leaves lingering questions and suggests we may still be missing important conformational changes, particularly given conformational dynamics observed spectroscopically in structures lining the inner Cl⁻ pathway[30,33].

Motivated by CLC-ec1's physiological function of mediating extreme acid tolerance within the mammalian stomach[4], we explored its pH-dependent conformational changes at even lower pH. Although only the extracellular face encounters acidity in vivo, symmetric pH conditions greatly simplify biophysical assays. Crucially, electrophysiological studies confirm that at symmetric pH 3 CLC-ec1 remains fully active, preserving a 2:1 Cl⁻/H⁺ coupling ratio[1] and exhibiting increased transport rates as the pH is lowered symmetrically from 4.5 to 3.0[35].

Here, we show that hydrogen–deuterium exchange mass spectrometry detects a sharp increase in CLC-ec1 conformational dynamics between pH 4.5 and 3.0, prompting structural analysis by single-particle cryo-electron microscopy (cryo-EM). The cryo-EM structures reveal subtle but consequential conformational changes between pH 4.0 and 3.0, with MD simulations of the pH 3.0 structure uncovering hydration pathways formed by water wires and capturing opening of the inner gate. Integrating results from structure, function, and simulation studies, we present a model to explain the bidirectional 2:1 Cl⁻/H⁺ stoichiometric exchange mechanism of CLC exchange transporters.

## Results

### Hydrogen–deuterium exchange mass spectrometry reveals two conformational transitions

To investigate pH-dependent changes in CLC-ec1 conformational dynamics, we first employed hydrogen–deuterium exchange mass spectrometry (HDX-MS). This technique offers an unbiased (label-free) spectrometric approach to revealing structural properties and

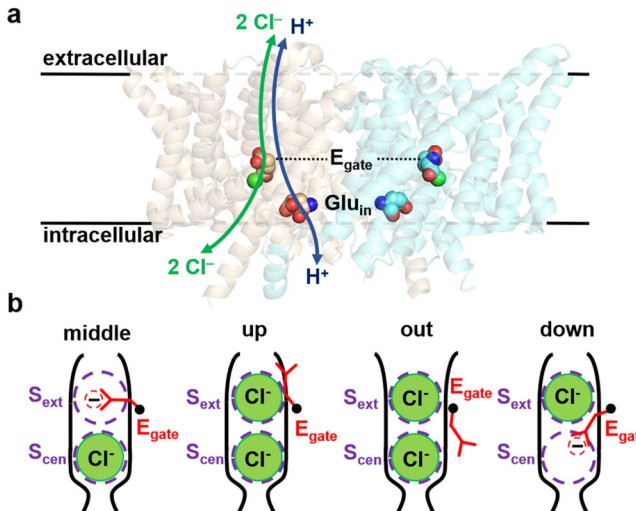

**Fig. 1 | CLC structure overview. a** Side view of CLC-ec1 (pdb 1OTS). The Cl⁻ and H⁺ permeation pathways are indicated by green and blue arrows, respectively. Bound Cl⁻ is shown as a green sphere. Key glutamate residues $E_{gate}$ and $Glu_{in}$ are shown space-filled. $E_{gate}$ physically gates the Cl⁻ permeation pathway and serves as an essential conduit for H⁺; $Glu_{in}$, though not strictly required, supports fast H⁺ transport along the intracellular portion of the H⁺ permeation pathway[10,74]. **b** Cartoon depictions of the Cl⁻ pathway, showing the conformations of $E_{gate}$ in high-resolution structures: "middle" (CLC-ec1[7]), "up" (CLC-ec1[9], CLC-7[75]), "out" (CLC-ec1[10,11]), CLC-1[76], and "down" (cmCLC[43], CLC-2[77]).

conformational dynamics in proteins[36]. It is especially advantageous for studying CLC-ec1 because it can identify subtle conformational changes in confined areas, such as the inner gate, without requiring a bulky label. HDX is based on the principle that structurally rigid, H-bonded protein regions incorporate deuterium into backbone amides slowly, whereas water-accessible dynamic protein regions do so rapidly.

CLC-ec1 was overexpressed in *E. coli* and purified using established methods[7,10,35] and exchanged into buffers spanning pH 3.0–6.5. Exchange was monitored across a time course from 20 to 63,000 s (17.5 h) at each pH. Optimized quench and digestion conditions yielded high sequence coverage (89%) with 237 peptides and strong redundancy (Supplementary Fig. 1). To confirm that the low-pH measurements reflect protein conformational dynamics rather than unfolding, we implemented two levels of control. First, within the HDX-MS workflow, we replicated the shortest labeling time point (20 s) using protein pre-incubated for 63,000 s in low-pH $H_2O$ buffer before labeling. Any significant structural perturbation would manifest as a difference in the measurement between these two conditions. Second, we analyzed the protein by size-exclusion chromatography (Supplementary Fig. 2), which verified that CLC-ec1 remained stable at low pH. These two controls support that the HDX measurements at low pH are not simply a result of protein unfolding or instability.

Since HDX is pH-dependent, quantitative comparison of datasets across different pH values requires correction. Such correction is relatively straightforward in the upper range (pH 4.5–6.5), where intrinsic exchange rates follow an approximately linear dependence and decrease by about tenfold per pH unit. However, at lower pH, the relationship deviates from linearity, and simple extrapolations become unreliable. Approximate correction factors were obtained as described and utilized for pH normalization in Supplementary Fig. 3, part B. Although these corrections rest on reasonable assumptions, they should be applied with caution. Notably, even without applying pH corrections, our data reveal key aspects of the pH-dependent conformational change in CLC-ec1. This is because, chemically, deuteration slows as the pH is lowered. In contrast, CLC-ec1 exhibits the opposite behavior in many regions—faster kinetics and higher deuteration as pH decreased—indicating enhanced solvent accessibility consistent with a protein conformational change.

Figure 2a illustrates the effect of pH on CLC-ec1 deuteration levels at 20 and 63,000 s. These rainbow-colored structures provide visual depictions of how deuteration of CLC-ec1 increases as pH is lowered. A more detailed representation of the data can be found in the uptake plots shown in Supplementary Fig. 3. In these plots, peptides from disordered regions (N- and C-termini, e.g., 3–21 and 453–461) follow the expected trend of slower exchange at low pH, while deeply membrane-embedded segments (e.g., 41–50, 336–344) show minimal exchange at any pH. In contrast, peptides along intra- and extracellular surfaces and the transport pathway displayed distinct exchange patterns. One group (e.g., 17–25, 22–28) exchanged faster at pH 6.5 than at 4.5, indicating a protein conformational transition over this pH range. A second transition is observed at pH ≤3.5, where another group of peptides (e.g., 51–64, 100–116, 146–163, 202–208) exchanged faster.

The first protein conformational transition involves a cluster of helices on the intracellular side of the protein, adjacent to the subunit interface. In Fig. 2a, this appears as similar deuterium exchange levels at pH 6.5 and pH 4.0 in those regions, despite a 100-fold faster intrinsic chemical exchange rate at pH 6.5. To view this transition more sensitively, we created a pH-corrected heatmap showing the deuteration difference between pH 6.5 and 4.5, compared at time points adjusted to account for their differing exchange rates (Fig. 2b). These changes align with previous cryo-EM analyses at pH 7.5 and 4.5, which showed conformational changes consistent with enhanced water accessibility in these regions[11]. Figure 2c, d and Supplementary Fig. 3c shows representative HDX data mapped onto the CLC-ec1 structure.

The second transition encompasses a broad swath of the protein, characterized by broadly accelerated exchange between pH 3.5 and 3.0. We note that although water exchange is extensive, the uptake remains region-specific: the deepest hydrophobic core regions are still protected, consistent with localized dynamic changes rather than global unfolding—in line with the protein pH-stability controls described above. This transition is illustrated by peptides 110–116 (helix D) and 146–163 (helix F), which form the inner and outer gates of the $Cl^-$ pathway. These segments became substantially more deuterated (Fig. 2c), indicating protein mobility that may facilitate $Cl^-$ and $H^+$ movement in these regions at pH 3.0. In contrast, part of helix B and helices C, I, J, M, and N remained protected or showed little exchange (Supplementary Fig. 3), although coverage of C and J was incomplete. Together, these findings suggest the existence of two distinct pH-dependent conformational transitions in CLC-ec1, one of which had not been previously detected.

## Cryo-EM structures of CLC-ec1

To visualize these pH-dependent changes at high resolution, we determined cryo-EM structures of CLC-ec1 at pH 7.5, 4.0, and 3.0, as summarized in Fig. 3 and Supplementary Table 1. Notably, the $E_{gate}$ was well-resolved in all three structures, as evidenced in the density maps and by the Q-scores[37] (Supplementary Figs. 4–6). In the pH 7.5 structure (3.18 Å resolution), $E_{gate}$ occupies the $S_{ext}$ anion-binding site in the "middle" orientation (Fig. 3a, d and Supplementary Fig. 4). This structure resembles the original CLC-ec1 crystal structure (1OTS, 2.51 Å resolution[9]), with a C-α r.m.s.d. of 1.00 Å. In the pH 4.0 structure (3.21 Å resolution), we observed a mixture of $E_{gate}$ in both the "middle" and "out" orientations, suggesting a transitional phase (Fig. 3b, e and Supplementary Fig. 5). In contrast, at pH 3.0 (3.14 Å resolution), $E_{gate}$ is observed only in the "out" orientation (Fig. 3c, f and Supplementary Fig. 6). These results confirm the pH-dependent modulation of $E_{gate}$ conformations.

In the pH 4.0 cryo-EM structure, the primary conformational change compared to pH 7.5 is an outward shift of helices A and G-H-I, which opens access to the $H^+$ permeation pathway (Fig. 3g). These changes at pH 4.0 compared to pH 7.5 are consistent with the increased water accessibility observed by HDX at pH 4.0 (Supplementary Fig. 3) and were also seen in the published pH 4.5 cryo-EM structure of CLC-ec1 (PDB ID: 7RP5)[11]. Overall, our pH 4.0 structure closely resembles 7RP5, with large RMSDs occurring mostly in loop regions that show low Q-scores in our maps (Supplementary Fig. 7a). The low Q-scores reflect the intrinsic flexibility of these loop regions and contribute to greater uncertainty in model-to-model comparisons in these regions. The most notable difference between our pH 4.0 structure and 7RP5 is the orientation of helix A, which angles away from the membrane in our structure compared with 7RP5 (Supplementary Fig. 7b). We hypothesized that this difference stems from the membrane-mimicking lipid nanodiscs used in our study versus detergent DM in 7RP5. Supporting this hypothesis, our pH 4.0 structure in DM reveals a helix-A orientation like that in 7RP5 (Supplementary Fig. 7b). This orientation change is likely driven by the stark physicochemical differences between the nanodisc's flat lipid bilayer and the small, curved detergent micelle, causing the helix to adjust its tilt angle to optimize its hydrophobic interactions. Future work should therefore investigate the functional role of specific lipid headgroups and their interactions with helix A.

In the pH 3.0 cryo-EM structure, there is a subtle conformational change compared to pH 4.0, with an overall C-α r.m.s.d. of 0.92 Å. Notably, the pH 3.0 conformation resembles the QQQ CLC-ec1 crystal structure (6V2J) (overall C-α r.m.s.d. 0.84 Å) more closely than the pH 4.0 structure (overall C-α r.m.s.d. 1.27 Å) (Fig. 3h). Compared with the pH 4.0 structure (Fig. 3h) and with the pH 4.5 structure 7RP5 (Supplementary Fig. 7c), the pH 3.0 map indicates a more dynamic conformational landscape: the H-I loop shows high flexibility, as reflected

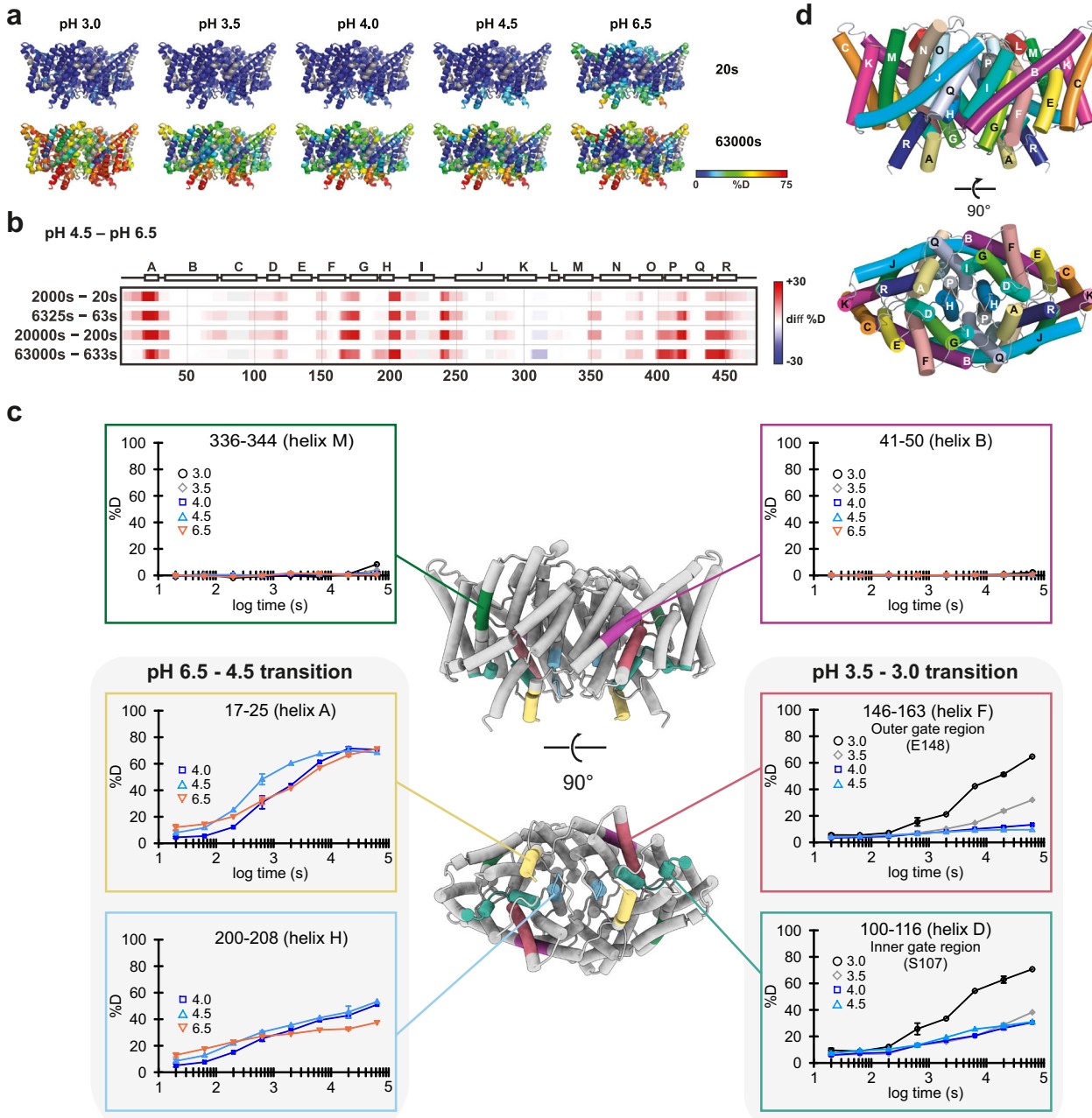

**Fig. 2 | HDX-MS analysis of CLC-ec1 reveals low pH-driven dynamics. a** CLC-ec1 structure (PDB: 1OTS) rainbow-colored according to back-exchange corrected deuteration levels at the beginning (20 s) and end (63,000 s) of the labeling experiment under different pH conditions. These views are presented without pH correction. Visually, these simple representations illustrate how lowering the pH increases deuteration, a trend opposite to the expected effect on chemical exchange, indicating that the observed changes arise from changes in protein dynamics. **b** pH-corrected heatmap to more sensitively visualize the change in protein dynamics occurring between pH 6.5 and 4.5, because the intrinsic deuterium-exchange rate at pH 4.5 is 100-fold slower than at pH 6.5, comparison of time points 100-fold different (e.g., 20 s at pH 6.5 vs. 2000 s at pH 4.5) reveals differences in exchange that are due to differences in protein dynamics. **c** Selected uptake plots of representative peptides were mapped by color onto the CLC-ec1 structure (views from the membrane plane and from the intracellular side). Standard deviations are shown for replicated time points (20 s ($n = 6$), 633 s ($n = 3$), 20,000 s, $n = 3$). Plots at the top represent regions undergoing little or no change, whereas those at the sides depict peptides capturing major pH-driven transitions. The uptake data were not corrected for pH-dependent differences in intrinsic exchange rates, and therefore, the curves are not directly comparable. Nevertheless, the common trend of increased dynamics (manifested as higher deuteration/faster kinetics) at lower pH illustrates the structural changes in CLC-ec1. The complete set of uptake plots is available in Supplementary Fig. 3, which also contains a set of plots where the differences in intrinsic exchange rates were compensated for by applying theoretically calculated correction factors (parts B and C). **d** Membrane and intracellular views of CLC-ec1 with helices labeled.

by low Q scores, and Helix A is absent. These cryo-EM features agree with our HDX results and with the multiple heterogeneous particle classes seen in the pH 3 dataset (Supplementary Fig. 6a). Although many regions are flexible at pH 3.0, the conserved pore region remains well-resolved. The pore radius profiles of the pH 3.0 and 4.0 structures

are similar (Fig. 3i), but at pH 3.0 $E_{gate}$ is shifted towards the pore and F357 adopts the "up" position, which in simulations lowers the energy barrier for Cl⁻ movement[24] (Fig. 3j). These subtle conformational changes at pH 3.0 vs 4.0 motivated us to perform MD simulations on the pH 3.0 structure.

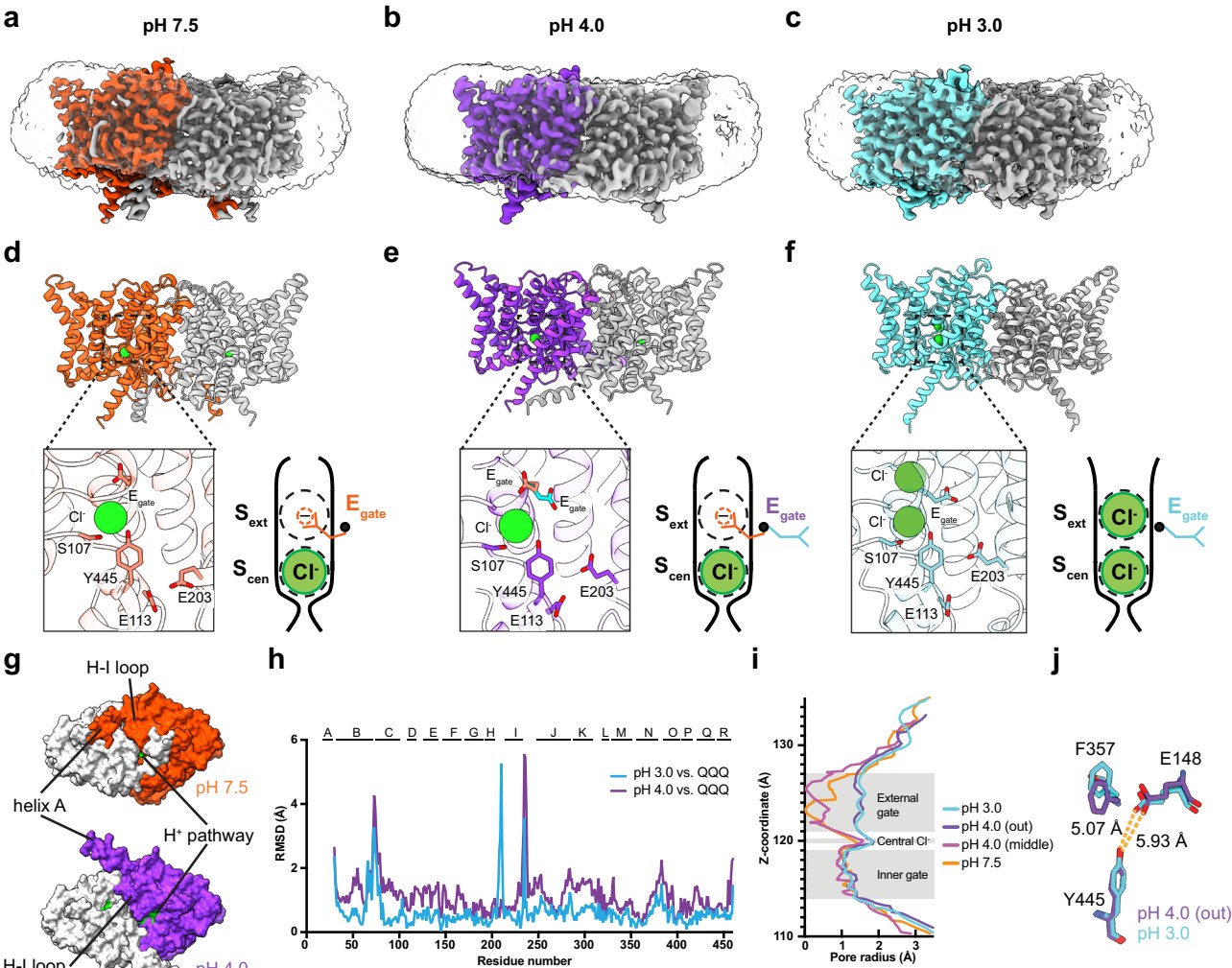

**Fig. 3 | Cryo-EM structures of CLC-ec1. a–c** Cryo-EM density maps for CLC-ec1, with each panel showing one of the two subunits in color: pH 7.5, orange; pH 4.0, purple; pH 3.0, cyan. **d–f** Molecular models of CLC-ec1 for structure determination at pH 7.5, 4.0, and 3.0, with colors as in (**a–c**). The zoomed-in views of the models showing key residues and resolved Cl⁻ ions. Cartoons depictions to the right of the zoomed-in views are as in Fig. 1, highlighting the E$_{gate}$ conformations observed at pH 7.5 ("middle" conformation), pH 4 ("middle" and "out" conformations), and pH 3 ("out" conformation). **g** View from the intracellular side, highlighting major conformational changes between pH 7.5 and 4.0. Residues lining the H⁺ pathway are colored in green (E113, E148, A182, L186, A189, F190, F199, E202, E203, M204, I402,

T407, and Y445), to highlight the opening of this pathway in the pH 4.0 conformation. **h** Comparison of pH 4.0 and pH 3.0 cryo-EM structures to the QQQ crystal structure. The structural alignment was done in ChimeraX using the matchmaker command, aligning the dimer. The high RMSD at the H–I loop reflects greater flexibility in the pH 3.0 structure. At pH 7.0 (QQQ), Q207 interacts with E117, stabilizing the loop. Protonation of E117 at pH 3.0 likely disrupts this interaction, increasing loop mobility. **i** Pore radius profiles calculated using HOLE. **j** Zoomed-in view comparing E148 and F357 between pH 4 (out) and pH 3. The inner gate residue Y445 is shown for reference.

## MD simulations reveal water wires, Cl⁻ leaving, and inner-gate opening

Water wires (chains of hydrogen-bonded water molecules) can facilitate proton conduction via a Grotthuss mechanism[38,39]. Early simulations on CLC-ec1 showed water wires transiently connecting E$_{gate}$ to Glu$_{in}$[18–25], which is a residue near the intracellular side of the transmembrane domain (Fig. 1a) whose neutralization has a major impact on H⁺ transport[10,17]. More recently, water wires connecting E$_{gate}$ directly to the intracellular solution were observed in MD simulations of the QQQ mutant crystal structure[10] and in simulations of WT CLC-ec1 pH 4.5 cryo-EM structure[11]. These water wires provide a plausible explanation for the transfer of H⁺ to and from E$_{gate}$ when it is in the "out" position. However, they do not account for all H⁺ transport steps needed to model a complete, reversible Cl⁻/H⁺ transport cycle, including how E$_{gate}$ becomes protonated when it is in the "down" position within the anion pathway. Simulations of the pH 3.0 cryo-EM

structure could provide further insights into this question or shed light on other aspects of the transport cycle.

We performed extended molecular dynamics simulations on the pH 3.0 cryo-EM structure in order to evaluate protein dynamics and water-wire formation. We observed three classes of water wires. Two classes (wires 2 and 3) resembled water wires seen in previous simulations, connecting the intracellular solution to E$_{gate}$ in the "out" position, whereas the third class (wire 1) was a water wire not observed in prior simulations, connecting the intracellular solution to the anion permeation pathway (Fig. 4a).

Water wires 2 and 3 offer a plausible explanation for how a proton moving to the inside is coupled to two chloride ions moving out, as illustrated in Fig. 4b. Glu$_{in}$ (Fig. 1a) is not depicted in this diagram because, although it substantially enhances H⁺ transport rates[17], it is not strictly required for H⁺ transport[10]. Supplementary Fig. 8 shows examples of water wires that are bordered by Glu$_{in}$ as well as those that

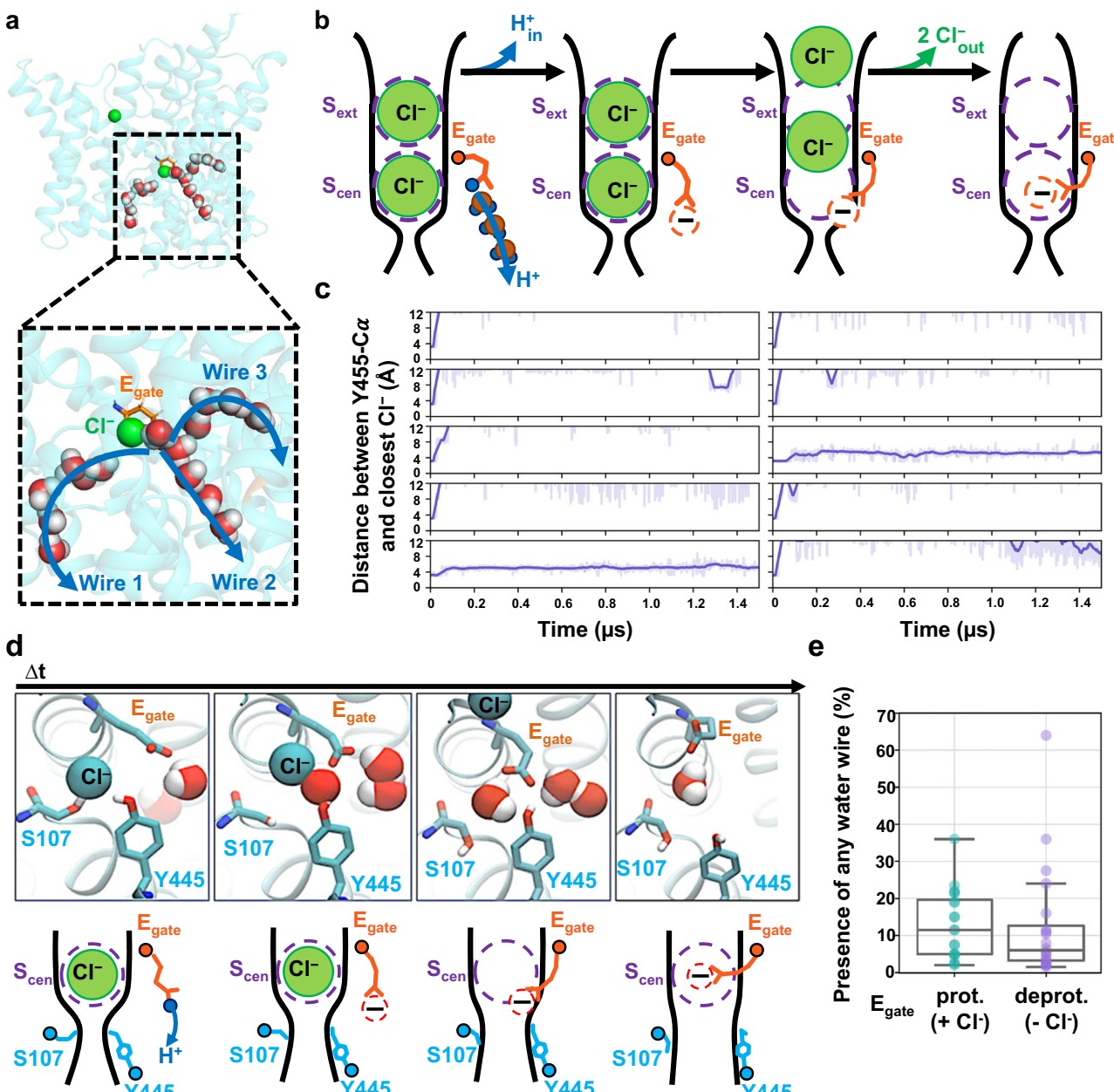

**Fig. 4 | Molecular dynamics simulations reveal water wires, Cl⁻ leaving, and inner-gate opening. a** Simulation snapshot depicting water wires that connect to the intracellular solution. **b** Cartoon model of proton transfer from $E_{gate}$ to the intracellular solution along water wire 2. The resulting negatively charged $E_{gate}$ is predicted to return to the anion pathway, expelling two Cl⁻ ions. **c** Cl⁻ leaves during eight out of ten 1.5-μs simulations with $E_{gate}$ deprotonated. Each plot shows the distance from inner-gate residue Y445 to the nearest Cl⁻ ion; >12 Å indicates that Cl⁻ no longer occupies anion-binding sites with the pathway. Unsmoothed traces (light purple lines) and traces smoothed with a moving average (dark purple lines) are shown for all simulations. The time traces were smoothed using a moving average with a window size of 20 ns. **d** Simulation snapshots showing a representative Cl⁻-leaving event. Concomitant with Cl⁻ leaving to the extracellular side (i.e., away from the viewer), inner-gate residues S107 and Y445 move away from one another,

opening the inner gate. Cartoon depictions are shown beneath each snapshot. **e** Water wires occurred with similar abundance in $E_{gate}$ protonated and deprotonated simulations, indicating that Cl⁻ binding does not substantially influence water-wire formation. Each dot corresponds to one simulation and shows the fraction of frames in which any of the three classes of water wires is formed. The middle line of each box in the plot is the median across simulations, with the box extending from the 1st to the 3rd quartile and defining the interquartile range. Whiskers extend to the last data points that are within 150% of the interquartile range. Note that Cl⁻ is always bound in simulations with $E_{gate}$ protonated, whereas no Cl⁻ is bound in over 70% of simulation frames with $E_{gate}$ deprotonated. A simulation snapshot showing water wires in the absence of Cl⁻ is shown in Supplementary Fig. 10.

are not. Following the delivery of a proton from the "out" $E_{gate}$ to the intracellular solution, the resulting negatively charged $E_{gate}$ will be energetically favored to move from the hydrophobic core of the protein to the anion-friendly pore. To achieve this, $E_{gate}$ must displace any Cl⁻ ions occupying the pore. Since CLCs are antiporters with a strict 2:1 Cl⁻/H⁺ exchange stoichiometry, the transport of a proton to the

intracellular solution would require the deprotonated $E_{gate}$ to displace 2 Cl⁻ ions to the outside (Fig. 4b).

Capturing the displacement of 2 Cl⁻ ions by $E_{gate}$ in simulations would provide strong support for this proposed water-wire mechanism. We therefore performed simulations on the CLC-ec1 pH 3.0 structure with $E_{gate}$ deprotonated and evaluated the presence of Cl⁻ in

the permeation pathway by plotting the distance from pore-lining inner-gate residue Y445 to the nearest Cl⁻ ion. Strikingly, in eight of ten independent 1.5-μs trajectories, both Cl⁻ ions left the permeation pathway (Fig. 4c). In stark contrast, Cl⁻ remained bound throughout all ten simulations with $E_{gate}$ protonated (Supplementary Fig. 9). Thus, the deprotonation of $E_{gate}$ is a crucial event that triggers expulsion of Cl⁻ from the permeation pathway. Inspection of the trajectories revealed that the Cl⁻ ions always left to the extracellular side, in harmony with the experimentally observed 2:1 Cl⁻/H⁺ exchange. Surprisingly, movement of $E_{gate}$ into the anion pathway was accompanied by reorientation of inner-gate residues S107 and Y445 (Supplementary Movie 1 and Fig. 4d).

### Water wires do not depend on chloride

The binding of Cl⁻ to the $S_{cen}$ site (Fig. 1b) is recognized as a critical feature in the CLC coupling mechanism and is supported by multiple lines of evidence. First, mutations that weaken Cl⁻ binding to $S_{cen}$ also reduce coupling efficiency, resulting in fewer H⁺ ions being transported per Cl⁻ ion[12,26]. Second, polyatomic anions that bind less strongly to $S_{cen}$ than Cl⁻ also exhibit weaker coupling to H⁺[27]. Finally, Cl⁻ binding at $S_{cen}$ supports formation of water wires that connect $Glu_{in}$ to $E_{gate}$, and mutations that disrupt these water wires reduce coupling efficiency[18,23,40]. Given these previous findings, we anticipated that the water wires connecting $E_{gate}$ to the intracellular solution would depend on Cl⁻ binding at $S_{cen}$. To test this hypothesis, we quantified water-wire formation in our two sets of simulations: with $E_{gate}$ protonated (Cl⁻ always present in the permeation pathway) and with $E_{gate}$ deprotonated (Cl⁻ absent during >70% of the simulation time). Surprisingly, we found that water wires were similarly abundant in both simulations (Fig. 4e and Supplementary Fig. 10), suggesting that water wires do not depend on Cl⁻ binding.

### Weak Cl⁻ binding at $S_{cen}$ does not necessarily lead to uncoupling

These simulation results inspired us to investigate the relationship between Cl⁻ binding and the coupling mechanism. Prior studies showed that any disruption to anion binding at $S_{cen}$ results in uncoupling[12,26,27]. As a result, it has been reasonably suggested that a key distinction between CLC transporters and channel homologs lies in their Cl⁻ binding affinity at $S_{cen}$, with CLC channels functioning as uncoupled channels rather than coupled transporters due to their reduced Cl⁻ binding affinity[13]. But does weak Cl⁻ binding inevitably result in uncoupling? The mutagenesis studies that established the link between weakened anion binding and uncoupling in CLC transporters involved the mutation of residues S107 and Y445. These residues not only directly interact with Cl⁻ at $S_{cen}$, but they also directly form the inner gate that controls Cl⁻ passage to and from $S_{cen}$ (Fig. 4d). Consequently, mutations at these positions physically alter the inner gate, and it is therefore possible that the uncoupling caused by these mutations is more a result of gate disruption than of weakened Cl⁻ binding.

To test the hypothesis that weak Cl⁻ binding alone does not cause uncoupling in the CLC transporter, we needed to disrupt Cl⁻ binding while leaving the inner gate intact. To do so, we focused on K131, as free energy calculations identified it as the most influential side chain in stabilizing Cl⁻ binding at $S_{cen}$[41]. K131 does not directly interact with Cl⁻; rather, the positively charged side chain is buried within the transmembrane domain, with the positively charged side chain pointing toward $S_{cen}$ and stabilizing Cl⁻ binding from a distance of 7–9 Å from $S_{cen}$[41]. We first experimentally confirmed weak Cl⁻ binding by K131A CLC-ec1, as well as the overall structural integrity of this mutant. The cryo-EM structure of K131A CLC-ec1 exhibited minimal differences in overall structure compared with WT CLC-ec1 (overall C-α r.m.s.d. 0.52 Å), except for a clear reduction of Cl⁻ density at $S_{cen}$ (Fig. 5a and Supplementary Fig. 11). Isothermal titration calorimetry (ITC) confirmed weak Cl⁻ binding by CLC-ec1 K131A (Fig. 5b), with the lack of heat detection indicating $K_d > 20$ mM[12].

To evaluate the transport function of CLC-ec1 K131 mutants, we used quantitative ion flux assays on purified transporters reconstituted into phospholipid vesicles[42] (Fig. 5c). We examined both K131A (for which we have structural and ITC data) and K131M, which was previously studied using planar lipid bilayer recordings[17]. We found that the unitary Cl⁻ turnover rate (Cl⁻ ions/s per transporter) was greatly reduced in K131 mutant transporters compared to WT (Figs. 5d, e and 1). Their unitary H⁺ turnover rates were also greatly reduced, to levels just measurable above background (Fig. 5d,e and Table 1). Importantly, we structured this flux-assay experiment to ensure that any H⁺ movement into the vesicles was a result of Cl⁻-dependent H⁺ pumping rather than leakage: by imposing a twofold gradient for H⁺ on the vesicles, any leak will involve movement of H⁺ out of the vesicles, and any H⁺ movement into the vesicles must occur via CLC-ec1 actively transporting H⁺ (Fig. 5c). Therefore, the low H⁺ transport rates measured for K131 mutant transporters represent bona fide H⁺ pumping rates. By analyzing the ratio of the Cl⁻ and H⁺ transport rates, we determined the Cl⁻/H⁺ transport stoichiometry (and thus the strength of coupling) for WT and mutant CLC-ec1 transporters. Strikingly, coupling in the K131 mutant transporters was largely intact, with just a slight increase in Cl⁻/H⁺ transport stoichiometry from ~2 for WT to ~3 for both K131 mutants (Fig. 5f and Table 1). This modest reduction in Cl⁻/H⁺ coupling efficiency is consistent with findings from K131M studied in planar lipid bilayers, where shifts in reversal potentials suggested slight uncoupling[17]. Together, these results illustrate that weak binding at $S_{cen}$ does not on its own cause uncoupling.

The dramatic slowdown in transport rates by K131A and K131M mutant transporters (Fig. 5d, e) could result directly from weakened Cl⁻ affinity, with Cl⁻ binding being rate-limiting for transport. Alternatively, reduced Cl⁻ occupancy could indirectly hinder transport by causing the K131 transporters to stall at one or more steps in the coupled transport cycle. To distinguish these possibilities, we evaluated the effect of K131 mutations in the background of the uncoupled mutant E148A. Lacking a titratable residue at the critical $E_{gate}$ position, CLC-ec1 E148A is completely unable to transport H⁺ and operates strictly as a passive Cl⁻ transporter[1]. If K131 mutations slow transport as a direct result of weakened Cl⁻ binding, then they should affect Cl⁻ transport rates in both WT and E148A backgrounds. On the other hand, if K131A mutations slow transport because they stall one of the $E_{gate}$ steps in the coupled transport cycle (either protonation/deprotonation of $E_{gate}$ or $E_{gate}$/Cl⁻ competition), then they should have no effect in the E148A background. We therefore tested CLC-ec1 K131A/E148A and CLC-ec1 K131M/E148A mutant transporters using the quantitative Cl⁻ flux assay (Fig. 5c), measuring only Cl⁻ since E148 transporters do not transport H⁺. K131 mutations had no discernible effect on E148A Cl⁻ transport rates (Fig. 5g and Table 1). These results align with the stalled-transport-cycle model: K131 mutations impair an $E_{gate}$-dependent step in the Cl⁻/H⁺ coupled transport cycle; when $E_{gate}$ is inoperative (E148A), Cl⁻ transport rates are unaffected.

## Discussion

A comprehensive model to explain the CLC transporters' 2:1 Cl⁻/H⁺ exchange mechanism has been elusive. This study was conducted with the hypothesis that investigating CLC conformational dynamics across a broader expanded pH range— encompassing the physiological pH associated with maximal activity—would yield insights into the exchange mechanism.

### Conformational dynamics at pH 3.0

Previous spectroscopic and cryo-EM studies revealed that the CLC-ec1 transporter undergoes conformational change at pH 4.5 in comparison to pH 7.5 and that this conformational change is essential for activity[11,16]. Our HDX-MS results align with those previous results and reveal additional conformational dynamics that occur at lower pH values, down to pH 3.0. Our cryo-EM structure at pH 4.0 reveals a

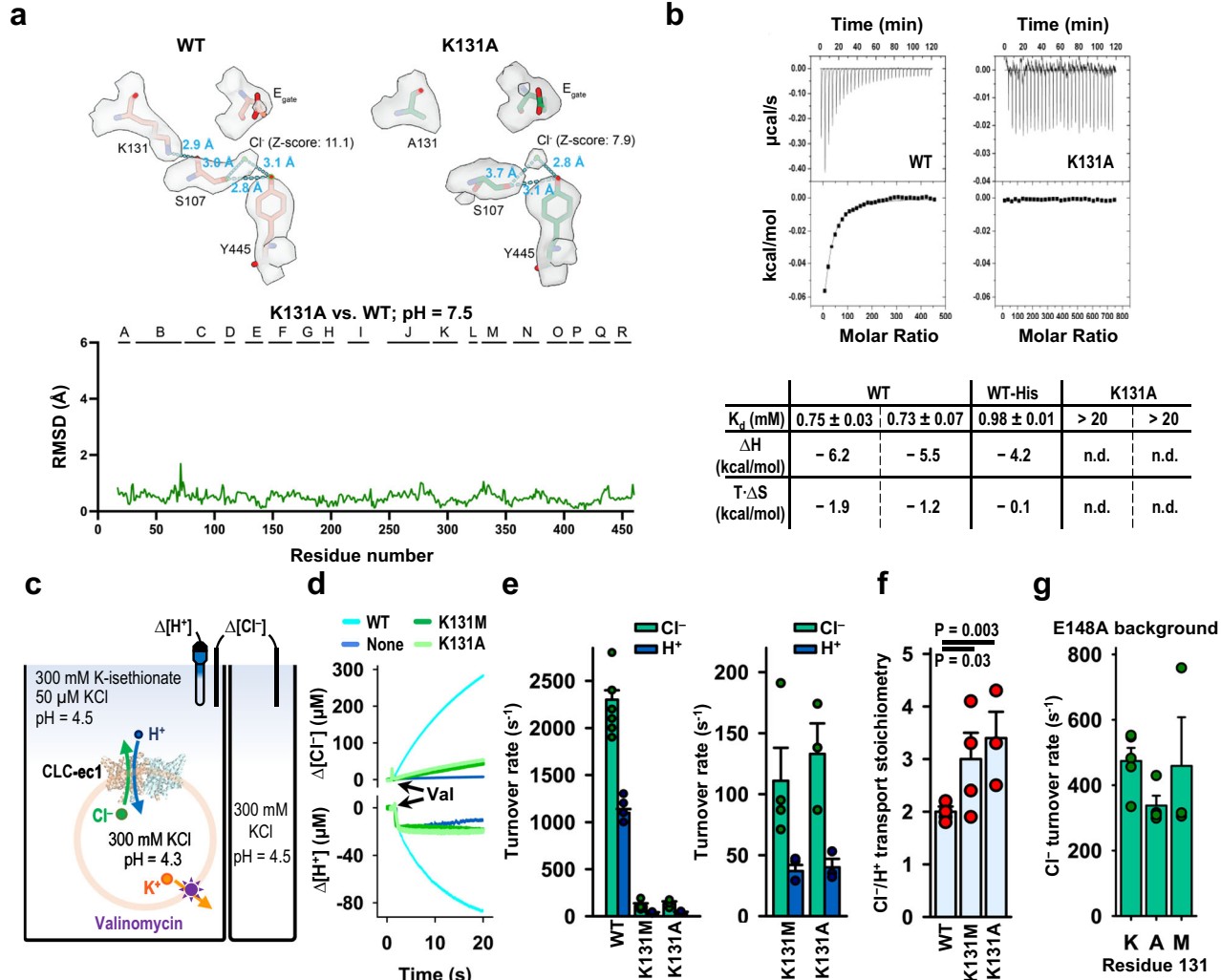

**Fig. 5 | K131 mutation reduces Cl⁻ binding but maintains Cl⁻/H⁺ coupling.**
**a** Comparison of K131A and WT CLC-ec1 cryo-EM structures. The Cl⁻ signal at $S_{cen}$ is substantially less distinct in K131 compared to WT (7.9 vs 11.1 sigma z-score), while the overall structures are highly similar (RMSD = 0.52 Å). Structural alignment was done in ChimeraX using the matchmaker command, aligning the dimer. **b** Cl⁻ binding measured by ITC for WT and K131A CLC-ec1, showing representative primary data and a summary table for experiments performed on 3 (WT) or 2 (K131A) separate protein preparations. For K131, no binding enthalpy was detected (n.d.) in either preparation. **c** Cartoon depiction of the functional assay to quantify H⁺/Cl⁻ transport rates and coupling stoichiometry. Extravesicular [Cl⁻] and [H⁺] were simultaneously measured. Because the Cl⁻/H⁺ exchange cycle nets three charges across the membrane per cycle, the buildup of an electrical gradient means that measurable Cl⁻ or H⁺ transport does not occur until the addition of valinomycin, which dissipates the electrical gradient by

shuttling K⁺ ions. The experimental setup involves a twofold gradient for H⁺, such that any leak will involve movement of H⁺ out of the vesicles, and any H⁺ movement into the vesicles must occur via CLC-ec1 actively transporting H⁺ via coupling to the downhill movement of Cl⁻. **d** Representative Cl⁻ and H⁺ traces for flux assays. "Val" indicates the time of addition of valinomycin. The signals indicate net Cl⁻ release from vesicles and net H⁺ uptake into the vesicles. **e** Cl⁻ and H⁺ transport rates (mean ± SEM) for WT and K131 mutant CLC-ec1, shown in two plots using different y axis scales: one optimized for comparison to WT, the other for visualizing mutant values more clearly. **f** Cl⁻/H⁺ coupling stoichiometry (mean ± SEM) of WT and K131 mutant transporters. P values were determined using a one-sided Student's t test. **g** Summary of transport rates (mean ± SEM) on uncoupled transporters (E148A background) with K (WT), A, or M at the 131 position. For (**e**–**g**), individual values of mean, SEM, and n (biological replicates) are presented in Table 1.

conformational change that opens access to the H⁺ permeation pathway (Fig. 3g), similar to that reported in cryo-EM studies of CLC-ec1 at pH 4.5[11]. At pH 3.0, we observe further conformational changes, particularly involving residues near the ion permeation pathways (Fig. 3h, j).

In a recent cryo-EM study on CLC-ec1 by Fortea et al., a "Twist" conformation featuring a notable 28.4-degree rotation between subunits was resolved at pH 4.5[11]. Interestingly, we did not observe this conformation in our pH 4.0 dataset. Our initial hypothesis was that the difference might stem from the membrane mimetics used: we employed Salipro lipid nanodiscs, whereas Fortea et al. used the detergent decyl maltoside (DM). To test this hypothesis, we determined a cryo-EM structure under our original conditions (pH 4.0, 150 mM NaCl) but using DM instead of Salirpo nanodiscs. However, we

still did not detect the "Twist" conformation; thus, the difference in membrane environment does not account for our lack of detection of the Twist conformation. Other differences between our experimental preparation and that of Fortea et al. include pH (4.0 vs 4.5), NaCl concentration (150 mM vs 100 mM), and buffer (citrate vs acetate). Given that the Twist conformation is not essential for function (preventing its formation with cross-links has no effect on activity[11,14]), it is perhaps not surprising that it is not detected under all conditions that support activity.

### Conformational dynamics revealed by molecular dynamics simulations
Our molecular dynamics simulations on the pH 3.0 cryo-EM structure provided unprecedented visualization of key steps in the Cl⁻/H⁺

**Table 1 | Turnover rates and stoichiometry of the proteins in this study**

| CLC-ec1 | Transport rate (s⁻¹) | | Cl⁻/H⁺ stoichiometry | n |
|---|---|---|---|---|
| | Cl⁻ | H⁺ | | |
| WT | 2300 ± 100 | 1100 ± 40 | 2.0 ± 0.1 | 6 |
| K131A | 110 ± 30 | 40 ± 10 | 3.0 ± 0.5 | 4 |
| K131M | 130 ± 30 | 40 ± 10 | 3.4 ± 0.5 | 3 |
| E148A | 470 ± 40 | n/a | n/a | 5 |
| K131A/E148A | 340 ± 30 | n/a | n/a | 4 |
| K131M/E148A | 460 ± 150 | n/a | n/a | 3 |

Transport rates and stoichiometry are displayed as the mean ± SEM. The stoichiometry is calculated from the ratio of the Cl⁻ and H⁺ transport rates. The number of flux assays (biological replicates) for each experiment is shown by *n*. At least two independent protein preparations were used for each protein sample.

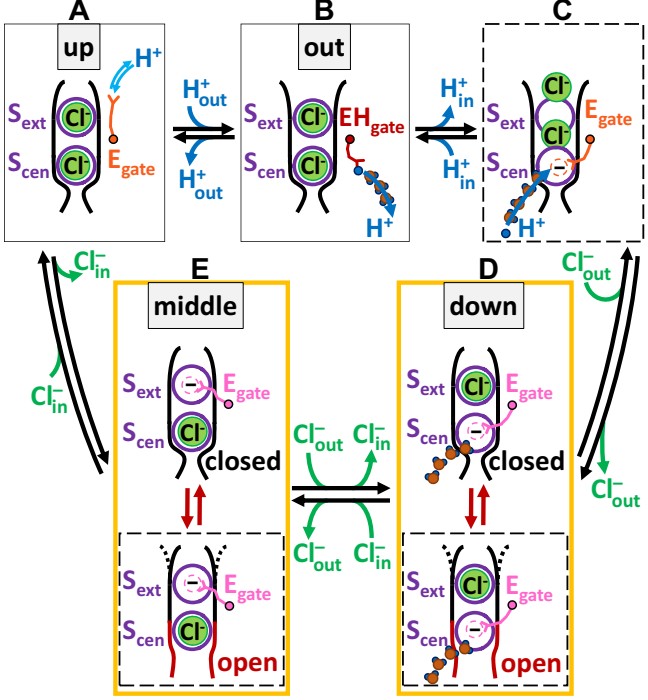

**Fig. 6 | Model for reversible Cl⁻/H⁺ exchange by CLC transporters.** The 2:1 stoichiometric transport mechanism involves conformational states **A-E** identified in structural studies (labeled "up", "out", "down", and "middle", shown in solid boxes), together with transient states inferred from simulations (shown in dashed boxes). For clarity, water wires are shown only at key H⁺-transport steps. The cycle, which can proceed in either direction, facilitates the transmembrane exchange of two Cl⁻ ions for one H⁺. A detailed description of the model can be found in the main text. All steps in this mechanism are inherently reversible. The relative rates of these steps and the net direction of transport will depend on ion concentrations, ion gradients, and the transmembrane voltage.

exchange mechanism. With E$_{gate}$ deprotonated, we dynamically observed the conformational change of E$_{gate}$ from the "out" to the "down" position. Remarkably, this conformational change occurred in concert with the displacement of two Cl⁻ ions into the extracellular solution, directly illustrating CLC-ec1's 2:1 exchange stoichiometry. In addition, the accompanying reorientation of inner-gate residues S107 and Y445 (Supplementary Movie 1 and Fig. 4d) was an unexpected observation that provides further clarity on the exchange mechanism. This reorientation unveils a long-awaited snapshot of the inward-open conformational state of CLC transporters. Since this conformation has

not been seen in any static structures, including structures in the absence of Cl⁻[26], we interpret that the simulations have captured a transient, functionally important state.

Our simulations revealed water wires that connect the intracellular solution to the CLC-ec1 protein core. These wires occur both through the canonical H⁺ pathway, as have been reported previously[10,11,18–25] and through the Cl⁻ pathway (Fig. 4a). Surprisingly, these water wires occur regardless of whether Cl⁻ is bound (Fig. 4e), contradicting prior simulations that indicated water wires depend on Cl⁻[18,23]. However, the prior simulations were conducted on a pH 7 structure now thought to represent an inactive or poorly active conformation[11]. Additionally, the water wires observed in prior simulations connected E$_{gate}$ only to Glu$_{in}$, which is beneficial but non-essential for H⁺ transport[10], and not to the intracellular solution. Nonetheless, the simulation result is surprising because the water wires that form in the absence of Cl⁻ appear to create a pathway for uncoupled transport. In this scenario, H⁺ transfer from the intracellular solution to E$_{gate}$ could be followed by movement of protonated E$_{gate}$ to the "out" position, allowing a proton to be transferred to the extracellular solution without concomitant Cl⁻ movement. However, the risk of uncoupled proton occurring in the Cl⁻-free transporter may be alleviated if the protonation energy for E$_{gate}$ is excessively high in the absence of Cl⁻. Indeed, free energy calculations conducted on CLC-ec1 in its low-activity state have predicted this phenomenon, with experimental ion-binding studies supporting the computational prediction[13]. Therefore, the observed abundance of water wires, regardless of the presence of Cl⁻, does not inherently pose a risk to coupling.

### The mechanism of reversible Cl⁻/H⁺ exchange

Our findings, combined with previous research, provide an unprecedented opportunity to propose a complete and efficient CLC 2:1 Cl⁻/H⁺ exchange mechanism, free from steps that are energetically unviable (Fig. 6). The transport mechanism involves conformational states identified in structural studies—"up," "out," "down," and "middle"—alongside transient states observed in simulations.

The cycle, which can proceed in either direction, facilitates the transmembrane exchange of two Cl⁻ ions for one H⁺, with each step characterized by energetically plausible mechanisms. Beginning at the top left with E$_{gate}$ in the "up" conformation (state A) and moving through the cycle clockwise, protonation from the extracellular solution enables the transition to the "out" conformation, with E$_{gate}$ in the hydrophobic core of the protein (state B). From there, water wires facilitate the transfer of a proton to the intracellular solution. (For simplicity of illustration, we show only one water wire, though multiple water wires may form.) Glu$_{in}$ (Fig. 1a) is not included in the model because it is non-essential in CLC-ec1[10] and is absent in some CLC transporters[43], indicating it is not required for the mechanism. Following H⁺ transfer to the intracellular side, the resulting negatively charged E$_{gate}$ moves from the hydrophobic core of the protein into the anion-friendly pore (transient state C), expelling Cl⁻ to the outside, either two ions at once, as observed in our simulations, or one at a time, as modeled here (states D and E). From the single-site occupancy states (D and E), which have been observed crystallographically ("down" and "middle"), the transient opening of the inner gate, as observed in our simulations, provides a pathway for Cl⁻ to enter from the intracellular solution. From state E, Cl⁻ entry expels E$_{gate}$ to the "up" conformation, allowing it to be protonated from the extracellular side, thus initiating another transport cycle.

Our simulations identified two key elements that shape the proposed transport mechanism: inner-gate opening and a water wire. We model inner-gate open states as transient states because inner-gate opening appears in simulation but has not been captured in low-energy structural snapshots. In the simulations described here, 2 Cl⁻ ions are expelled to the extracellular solution, leaving a pathway with no bound Cl⁻. Structural data show that single-Cl⁻ occupancy states are

stable[7,43], so we model them as states D and E. To represent the short-lived inner-gate openings seen in simulations, we also include transient, inner-gate-open variants of these states (dashed boxes) that would permit $Cl^-$ movement to and from the intracellular solution. Although the simulation did not discretely capture these single-occupancy states, this is reasonably explained by the challenges of observing $Cl^-$ entry in simulations. Ions already in the permeation pathway can easily overcome small energy barriers to dissociate from the binding site within a simulation timescale, while the diffusion of $Cl^-$ to the binding site is a longer timescale process not captured within the length of our simulations. The primary takeaway is that the simulations demonstrate the feasibility of the inner-gate open state and thus support its role in the transport mechanism.

The second key element from our simulations is the identified water wire (Wire 1 in Fig. 4, illustrated here in states C and D), which connects the intracellular solution to $E_{gate}$ in the "down" conformation and offers a physically plausible pathway for the transport cycle to proceed in the counterclockwise direction, with $Cl^-$ ions moving inward and $H^+$ ions moving outward. Starting at the top left in state A and moving counterclockwise, the negatively charged $E_{gate}$ moves into the $Cl^-$ pathway, displacing a $Cl^-$ ion to the intracellular side and generating state E. Subsequent entry of a $Cl^-$ ion from the extracellular side knocks $E_{gate}$ to the "down" position and displaces a second $Cl^-$ ion, generating state D. In this conformation, water-wire 1 is connected to $E_{gate}$, but protonation of $E_{gate}$ is not yet energetically favorable, with only a single $Cl^-$ ion present in the pathway. Upon entry of a second $Cl^-$ (state C), protonation of $E_{gate}$ becomes energetically favorable, enabling proton transfer via water-wire 1. Following protonation, the neutralized $E_{gate}$ is favored towards the "out" conformation (state B). A subsequent shift to the "up" conformation (state A) enables release of a proton to the extracellular solution, thereby completing the transport cycle in that direction.

K131 is universally conserved across the CLC family, including both transporter and channel homologs. This conservation, combined with K131's location—entirely buried within the transmembrane domain, which is rare for lysine residues[44]— suggests that it plays a crucial functional role. Indeed, mutations at this position in CLC channels shift voltage dependence and lead to myotonia[45–47]. In CLC-ec1, K131 mutations weaken $Cl^-$ binding affinity and reduce $Cl^-/H^+$ transport rates (Fig. 5 and Table 1). Results from experiments on $E_{gate}$ mutant transporters (Fig. 5g) indicate that the transport slowdown stems from inhibition at one or more of the $E_{gate}$-dependent steps. We propose that key bottlenecks occur during transitions where $Cl^-$ and $E_{gate}$ compete for binding to the $S_{ext}$ and $S_{cen}$ sites. Although K131 mutations likely reduce $E_{gate}$'s binding affinity at these sites, its physical tethering confers a competitive advantage over free $Cl^-$ ions. In the context of the exchange mechanism (Fig. 6), transitions D⇌E, D → C, and E → A will be slow, stalling the transport cycle. This model for stalled transport is consistent with the K131A structure, which shows that the cryo-EM density for $E_{gate}$ is less affected compared to that of $Cl^-$. The model also aligns with the largely preserved coupling seen in K131 mutants, where mild uncoupling raises the $Cl^-/H^+$ stoichiometry from 2 to ~3. A more severe loss of $E_{gate}$ affinity would allow freer $Cl^-$ movement and would drive $Cl^-/H^+$ stoichiometry higher, as seen in other mutants with low $Cl^-$ binding affinity[12]. The mild uncoupling in K131 mutants could result from loss of hydrogen bonds between K131 and inner-gate loop residues A104 and G106, slightly destabilizing the inner gate and allowing an extra $Cl^-$ ion to slip through roughly once per transport cycle (Supplementary Fig. 12).

The proposed transport mechanism (Fig. 6) is relevant to CLC transporters across all kingdoms, which share fundamental features including 2:1 anion/proton coupling and $E_{gate}$ as an essential coupling element[3]. The mechanism elucidates how transport occurs in the direction of $Cl^-$ ions moving inward across the membrane and $H^+$ ions moving outward, which is the direction of physiological relevance for CLC-ec1 in facilitating extreme acid tolerance[4] and for mammalian

CLCs in acidifying intracellular compartments[8,48]. An alternative to our proposed 2:1 mechanism was previously suggested through elegant kinetic modeling, showing that multiple distinct pathways can collectively yield the experimentally observed stoichiometry[49]. Because those models were built on the CLC-ec1 structures available at the time, which were recently proposed to represent an inactive conformation[11], extending this modeling with the additional structures now available could provide valuable insight into how ensembles of pathways may operate in CLC transporters. A point of debate in such extensions is the stability and relevance of the $E_{gate}$ "out" conformation: while one computational study questioned its stability at physiological pH[50], other work has argued for its relevance[10,11,24,28]. Although $E_{gate}$ "out" has been detected only under symmetric low-pH conditions, these conditions are experimentally more accessible than asymmetric pH and still preserve the hallmark 2:1 $Cl^-/H^+$ exchange ratio, supporting the view that the fundamental coupling mechanism remains intact and, by extension, that the findings are relevant under physiological asymmetric pH. Looking ahead, integration of complementary structural, computational, and kinetic modeling approaches will be valuable for shaping our picture of the 2:1 coupling mechanism, building on the insights advanced here.

Our model posits that proton transport occurs along connected water wires identified in our MD simulations. These contiguous hydrogen-bonded chains provide the structural and electrostatic continuity required for Grotthuss-like hopping through a hydrophobic protein, marking the minimal requirement for proton transport. In our simulations, we did not model an excess proton in the water wires, and we did not simulate explicit proton transport; these simplifications allowed us to focus computational effort on sampling structural dynamics and hydration patterns. In simulations of the CLC-ec1 high-pH conformation, inclusion of an excess proton was found to draw additional solvated water into the protein core and create broader water-wire structures compared to simulations without the excess proton[51]. Explicit proton-inclusive simulations[40,51,52] would be an appropriate next step to quantify proton hopping kinetics and free-energy barriers, building on the structural and functional insights established here. Beyond this, reactive MD computational studies will be important for validating the proposed mechanism, as its ability to capture protonation and deprotonation events at specific residues provides mechanistic detail for assessing the dynamic interplay between proton transfer and protein conformational changes, complementing experimental studies.

While our proposed mechanism addresses the fundamental $Cl^-/H^+$ exchange function, questions remain concerning how this core function is regulated in the various CLC transporters. For example, mammalian CLCs contain large cytoplasmic domains that mediate regulation by signaling lipids[48] and by nucleotides[53,54], but these regulatory mechanisms are only partially understood. Many prokaryotic homologs also feature such domains, although their functional roles have yet to be investigated. Gaining insight into how these regulatory mechanisms interact with the core $Cl^-/H^+$ exchange process will be crucial for fully understanding the diverse functions of CLC transporters in cellular physiology.

## Methods

### Mutagenesis
Mutations were introduced into the WT CLC-ec1 gene in pASK[6] using New England BioLabs Q5 Site-Directed Mutagenesis kit (New England Biolabs, Ipswich, MA, USA). Each mutation was subsequently verified with whole plasmid sequencing and analyzed using SnapGene® software 2023 (from Dotmatics; available at snapgene.com).

### Protein production and purification
CLC-ec1 with an N-terminal polyhistidine tag was transformed into BL21 *E. coli* cells and plated onto LB plates containing 100 µg/mL

ampicillin. The following morning, the colonies from 1 to 2 plates were scraped into 10 mL LB, added to 1 L of Terrific Broth in a 2.8-L baffled flask, and cultured at 37 °C with vigorous shaking (~220 RPM). When the culture reached 1.0 $OD_{600}$, it was induced with 0.2 mg/L anhydrotetracycline (added from a 0.2 mg/mL solution in dimethylformamide). The cultures were grown for an additional 3 h, then pelleted at $2246 \times g$ for 20 min and flash-frozen in liquid nitrogen for future purification.

For purification, CLC-ec1 pellets were extracted with 50 mM n-Decyl-α-D-Maltopyranoside (DM) (Anatrace, sol grade) for 2 h, then centrifuged at $25,759 \times g$ for 45 min. The protein was then isolated using a cobalt-affinity chromatography step (3 mL of 50% cobalt metal affinity resin slurry, Takara Bio USA, Inc., San Jose, CA, for each 1 L prep), eluting with cobalt wash buffer (200 mM NaCl, 20 mM DM (sol grade), 400 mM Imidazole, 40 mM Tris Cl, pH 7). The eluate was incubated with Endoproteinase Lys-C (Santa Cruz Biotechnology, Inc.), 0.17 U per L culture. The final purification by size-exclusion chromatography (SEC) (Enrich SEC 650 10 ×300 column, Bio-Rad Laboratories Pleasanton, CA) was carried out in SEC Buffer (10 mM NaHepes, 150 mM NaCl, 5 mM DM (anagrade), pH 7.5). Purified protein was concentrated using Amicon® Ultra-15 centrifugal filters with a 50-kD cut-off.

For single-particle cryo-EM experiments at pH 7.5 and 4.0, CLC-ec1 purified by SEC as above was reconstituted into Saposin-lipoprotein nanoparticle (Salipro)[55] at either pH 7.5 and pH 4. The reconstitution protocol, detailed below, was adapted from Chien et al.[56] with some modifications. For the pH 7.5 Salipro sample preparation, 0.5 mg of CLC-ec1 in DM at pH 7.5 was mixed with 1.35 mM of POPE, 450 μM of POPG, 0.2 % (w/v) of DDM, 120 μM of Saposin-A, and Gel-filtration buffer (GFB) (10 mM HEPES, 150 mM NaCl, pH 7.5) to a total volume of 2.5 mL. The mixture was incubated at 4 °C for 45 min. Then 50 % (w/v) of Amberlite XAD-2 (Millipore Sigma) was added to the mixture followed by 15 min incubation at 4 °C. The Amberlite XAD-2 beads were removed by centrifugation. The mixture was concentrated to 200 μL using a 10 kDa cut-off Amicon (Millipore Sigma). A final SEC step was carried out using Superdex 200 Increase (10/300) (Cytiva) equilibrated in GFB. The fractions corresponding to CLC-ec1 in Salipro were collected and concentrated using a 3-kDa cut-off Amicon concentrator (Millipore Sigma) to a final concentration of 2.2 mg/mL. For the pH 4.0 Salipro sample, 1.35 mM of POPE, 450 μM of POPG, 0.2% (w/v) of DDM, 120 μM of Saposin-A and the pH 4 buffer (1.8 mM Citric Acid, 8.2 mM Sodium Citrate, 150 mM NaCl, pH 4) were mixed first and incubated at 37 °C for 10 min. Then 1.5 mg of CLC-ec1 in DM at pH 7.5 was added to the mixture, making a total volume of 5 mL. The mixture was incubated at 37 °C for 5 min. The rest of the steps including Amberlite XAD-2 beads and gel-filtration are the same as for the Salipro pH 7.5 sample.

For cryo-EM of WT CLC-ec1 at pH 3, protein was generated using the method as described above, but with a modification to generate a final sample in the detergent lauryl maltose neopentyl glycol (LMNG). During the cobalt-affinity chromatography step, DM gradually replaced with LMNG (Anatrace), to a final concentration of 20 μM LMNG. The exchange was as follows 50:50, 25:75, 10:90, and finally 5:95 percent by volume DM to LMNG. The final SEC step was performed in pH 3 buffer, 8.2 mM Citric Acid, 1.8 mM sodium Citrate, 150 mM NaCl, and 20 μM LMNG. For cryo-EM of K131A, the LMNG protocol was used as described above, except the pH in the final SEC buffer was 7.5 (10 mM NaHEPES, 150 mM NaCl, 20 μM LMNG, pH 7.5).

### Hydrogen-deuterium exchange mass spectrometry

Detergent-solubilized CLC-ec1 was purified following the established protocol, including a final SEC step in 7.5 mM HEPES, 100 mM NaCl, and 1.8 mM DM (Anagrade) at pH 7.5. The purified protein was then buffer-exchanged into $H_2O$-based McIlvaine buffers of defined pH, prepared by mixing solution A (0.2 M $Na_2HPO_4$) and solution B (0.1 M

citric acid) at recommended ratios (pH 3.0−A 206 μl, B 795 μl; pH 3.5−A 304 μl, B 697 μl; pH 4.0−A 386 μl, B 615 μl; pH 4.5−A 454 μl, B 546 μl; pH 6.5−A 710 μl, B 290 μl). These pre-mixes were diluted fivefold into water, and NaCl (final 100 mM) and DM (final 1.8 mM) were added. Protein buffer exchange was performed using Zeba Spin 7k MWCO desalting columns (Thermo Scientific), and the protein was incubated under these conditions for 1 h. Exchange reactions were initiated by fivefold dilution into $D_2O$-based buffers of identical composition and pD. Protein concentration was 3.2 μM and temperature 21 °C. Aliquots were collected at 20, 63, 200, 633, 2000, 6325, 20,000, and 63,000 s under all pH conditions. The exchange time intervals were identical across all samples regardless of the pH. To ensure that all samples resided in exchange solutions for exactly the same time, sample collection was performed manually. Time points 20, 633, and 20,000 s were acquired in triplicate. In addition, protein integrity at low pH was assessed using a specific control: CLC-ec1 was incubated for 63,000 s (equal to the longest time point) in $H_2O$ buffer and subsequently subjected to a 20 s pulse in $D_2O$ buffer (also done in triplicate). In all cases, the exchange was stopped by adding phosphoric acid (mixing ratio 1:1) and freezing in liquid nitrogen. Different phosphoric acid concentrations matching the exchange buffer's pH and containing 1.8 mM DM were used (pH 6.5− 60 mM, pH 4.5−40 mM, pH 4.0− 35 mM, pH 3.5−30 mM, 3.0−25 mM). A fully deuterated control was prepared to correct for back-exchange, accounting for deuterium lost during sample handling, chromatography, and MS analysis.

Each sample was quickly thawed and injected onto a protease column (bed volume 70 μl) containing immobilized pepsin and nepenthesin-2[57]. Digestion was done at 19 °C and was driven by a flow of 0.4% formic acid in water delivered by an Agilent 1260 Infinity II Quaternary pump (Agilent Technologies, Waldbronn, Germany) at 200 μL.min⁻¹. Peptides were trapped on a guard column (SecurityGuard™ ULTRA Cartridge UHPLC Fully Porous Polar C18, 2.1 mm ID, Phenomenex, Torrance, CA) and desalted. Next, separation on an analytical column (Luna Omega Polar C18, 1.6 μm, 100 Å, 1.0 × 100 mm, Phenomenex, Torrance, CA) was done by an acetonitrile gradient (10%−45%; solvent A: 0.1% FA in water, solvent B: 0.1%FA, 2% water in ACN). Solvents were delivered by the Agilent 1290 Infinity II LC system pumping at 50 μL.min⁻¹. To minimize carry-over and remove the detergent, the protease column was washed by two consecutive injections of 4 M urea, 500 mM glycine HCl pH 2.3 and 5% acetonitrile, 5% isopropanol, 20% acetic acid in water[58]. The analytical column, together with the guard column, was washed with solvent B for 4 min and then with 70% methanol, 20% isopropanol, and 1% formic acid in water for 4 min. The LC setup was cooled to 2 °C to minimize the deuterium loss. The analytical column outlet was directly connected to an ESI source of a timsTOF Pro (Bruker Daltonics, Bremen, Germany) operating in MS mode with 1 Hz data acquisition. The data processing started with peak picking in Data Analysis 5.3 and exported to simple text files, which were then further handled in DeutEx 1.0 (Bruker Daltonics, Bremen, Germany). Data visualization was done using MSTools Data visualization was done using MSTools (https://peterslab.org/MSTools/) and PyMol 3.1 (Schrödinger). Identification of peptides arising from pepsin/nepenthesin-2 digestion was done using the same LCMS system, but the mass spectrometer operated in data-dependent MSMS mode with PASEF enabled. The data were searched using MASCOT (v. 2.7, Matrix Science, London, UK) against a database containing sequences of CLC-ec1, acid proteases, and cRAP.fasta (https://www.thegpm.org/crap/). The search parameters were: precursor tolerance 10 ppm, fragment ion tolerance 0.05 Da, decoy search enabled with FDR <1%, IonScore > 20, and peptide length >5. The mass spectrometry data have been deposited to the ProteomeXchange Consortium via the PRIDE partner repository with the dataset identifier PXD058693.

For the stability control experiment, purified CLC-ec1 in 20 mM HEPES (pH 7.5), 100 mM NaCl, and 1.8 mM DM was transferred to

McIlvaine buffer (pH 3.0) containing 100 mM NaCl and 1.8 mM DM using Zeba Spin 7 K MWCO desalting columns (Thermo Scientific). After buffer exchange, the protein was incubated for 0, 1, 4, and 8 h before separation on a size-exclusion chromatography (SEC) column (Enrich™ SEC 650, 10 × 300 mm, 24 mL; Bio-Rad). ClC-ec1 maintained at pH 7.5 served as a control. SEC separation was performed using a 1 mL sample containing 1 mg of protein at a flow rate of 1 mL/min. The elution buffer consisted of 20 mM HEPES (pH 7.5), 100 mM NaCl, and 1.8 mM DM. All samples were analyzed under identical conditions.

### Cryo-EM sample preparation, data collection, and image processing

Purified samples with a concentration of 2–4 mg/mL were used for cryo-EM grid preparation. 1 mM fluorinated FOS-choline-8 was added prior to freezing grids to improve ice quality and avoid preferred orientation due to air-water interface interactions. In total, 3 μL of the sample was applied to glow-discharged Quantifoil R1.2/1.3 Cu200 grids, then blotted with a Whatman filter paper for 3 s before plunge-frozen in liquid ethane using a Vitrobot Mark IV (Thermo Fisher Scientific) at 4 °C and 100% humidity.

All cryo-EM data were collected with Thermo Fisher Scientific Titan-Krios cryo-electron microscopes operating at 300 keV. The WT pH 7.5 dataset was collected with a Falcon 4 camera without an energy filter. The WT pH 4 dataset and K131A dataset were collected with a Gatan K3 camera and Bio-quantum energy filter set to 20 eV. The WT pH 3 dataset was collected with a Falcon 4 and Selectris energy filter set to 10 eV. Data collection parameters are reported in Supplementary Table 1. Automated data collection was done using EPU v2.10 software (Thermo Fisher Scientific).

The complete data processing workflows with specific details for each sample/dataset are reported in Supplementary Figs. 4–6. The overall data processing strategy is similar for all four datasets (WT at pH 7.5, 4, and 3, and K131 at pH 7.5). The raw movies were pre-processed (motion correction and CTF correction) in CryoSPARC live v4.6 (Structura Biotechnology Inc). Micrographs were curated with defined criteria, including CTF fit and relative ice thickness. A blob picker or a template picker was used to pick a small subset of the dataset. The picked particles were pruned by 2D classifications, ab initio reconstructions, and heterogeneous refinements in cryoSPARC v4.6 (Structura Biotechnology Inc.). The good particles that could be reconstructed into a map with recognizable protein features were used to train a Topaz particle picker[59]. This Topaz picker was used to pick the full dataset. Several rounds of ab initio reconstructions and heterogeneous refinements with three classes were used to clean the particle stacks. For WT pH 7.5 and pH 3 datasets, 3D classification without alignment in Relion v4[60] with a mask that covers only the proteins was also used for particle cleanup. The final non-uniform refinement with C2 symmetry imposed was done in cryoSPARC v4.6. All the plots derived from cryo-EM data were prepared using Graphpad Prism v10.

### Model fitting, refinement, and analysis

A ClC-ec1 crystal structure (PDB: 1OTS) was used as an initial atomic model for WT ClC-ec at pH 7.5 and pH 4 and for K131A. The QQQ crystal structure (PDB: 6V2J) was used for pH 3 model building. The initial models were first modified based on our construct (i.e., K131A or WT sequences). These models were rigid-body docked into the cryo-EM map and refined using ISOLDE v1.6[61] and PHENIX v2.0[62]. All atomic models were validated in PHENIX validation job and Q-score[37]. All structure figures were prepared with UCSF ChimeraX v1.8[63].

To compare Cl⁻ densities between WT and K131A mutant structures, we calculated z-scores. The Z-score normalizes the map density by expressing it as the number of standard deviations it lies above the average density of the entire cryo-EM map. This calculation is:

$$Z = \frac{Density\ Value - Mean\ Density\ Value}{Standard\ Deviation\ of\ Density\ Value} \quad (1)$$

By performing this normalization, the z-score as a normalized contour level provides a statistically robust measure of how confidently the Cl⁻ ion signal (e.g., 11.1) can be distinguished from the background noise, enabling a fair comparison between the WT and mutant maps when their absolute intensity scales are different.

### System setup for molecular dynamics simulations

We performed simulations of ClC-ec1 under two conditions: (1) simulations with E148 ($E_{gate}$), E203 ($Glu_{in}$), and E113 protonated; (2) simulations with E148 ($E_{gate}$) deprotonated, and E203 ($Glu_{in}$) and E113 protonated. We initiated all simulations from a ClC-ec1 structure based on the cryo-EM data reported in this manuscript (specifically, from a model very similar to that presented here but based on an earlier refinement). The three Cl⁻ ions bound at $S_{cen}$, $S_{ext}$, and $S_{int}$ anion-binding sites in each of the two subunits were preserved. For each simulation condition, we performed ten independent simulations, each 1.5 μs in length. For each simulation, initial atom velocities were assigned randomly and independently.

For all simulation conditions, the protein structure was aligned to the Orientations of Proteins in Membranes[64] entry for 6V2J (ClC-ec1 triple mutant (E113Q, E148Q, E203Q)[10] using PyMOL 2.0 (Schrödinger), and crystal waters from 6V2J were incorporated. Prime (Schrödinger, version 2022-1)[65] was used to add capping groups to protein chain termini. Protonation states of all titratable residues except for E148 ($E_{gate}$), E203 ($Glu_{in}$), and E113 were assigned at pH 4.5, which is the pH at which activity is most often measured (e.g., Fig. 5). Using Dabble[66], version 2.6.3, the prepared protein structures were inserted into a pre-equilibrated palmitoyl-oleoyl-phosphatidylcholine (POPC) bilayer, the system was solvated, and sodium and chloride ions were added to neutralize the system and to obtain a final concentration of 150 mM. Simulations were conducted in the absence of an applied transmembrane voltage. The final systems comprised approximately 123,000 atoms, and system dimensions were approximately 160 x 120 x 90 Å (Supplementary Table 2). The reliability and reproducibility checklist for MD simulations is presented in Supplementary Table 3.

### Molecular dynamics simulation and analysis protocols

We used the CHARMM36m force field for proteins, the CHARMM36 force field for lipids and ions, and the TIP3P model for waters[67-69]. All simulations were performed using the Compute Unified Device Architecture (CUDA) version of particle-mesh Ewald molecular dynamics (PMEMD) in AMBER20 on graphics processing units (GPUs).

Systems were first minimized using three rounds of minimization, each consisting of 500 cycles of steepest descent followed by 500 cycles of conjugate gradient optimization. In all, 10.0 and 5.0 kcal·mol⁻¹·Å⁻² harmonic restraints were applied to the protein and lipids for the first and second rounds of minimization, respectively. 1 kcal·mol⁻¹·Å⁻² harmonic restraints were applied to the protein for the third round of minimization. Systems were then heated from 0 K to 100 K in the NVT ensemble over 12.5 ps and then from 100 K to 310 K in the NPT ensemble over 125 ps, using 10.0 kcal·mol⁻¹·Å⁻² harmonic restraints applied to protein heavy atoms. Subsequently, systems were equilibrated at 310 K and 1 bar in the NPT ensemble, with harmonic restraints on the protein non-hydrogen atoms tapered off by 1.0 kcal·mol⁻¹·Å⁻² starting at 5.0 kcal·mol⁻¹·Å⁻² in a stepwise fashion every 2 ns for 10 ns, and then by 0.1 kcal·mol⁻¹·Å⁻² every 2 ns for 20 ns. Production simulations were performed without restraints at 310 K and 1 bar in the NPT ensemble using the Langevin thermostat and the

Monte Carlo barostat, and using a timestep of 4.0 fs with hydrogen mass repartitioning[70]. Bond lengths were constrained using the SHAKE algorithm[71]. Non-bonded interactions were cut-off at 9.0 Å, and long-range electrostatic interactions were calculated using the particle-mesh Ewald (PME) method with an Ewald coefficient of -0.31 Å$^{-1}$, and 4th order B-splines. The PME grid size was chosen such that the width of a grid cell was ~1 Å. Trajectory frames were saved every 200 ps during the production simulations. The AmberTools17 CPPTRAJ package was used to reimage trajectories[72]. Simulations were visualized and analyzed using Visual Molecular Dynamics (VMD) 1.9.4[73] and PyMOL 2.0 (Schrödinger).

### Isothermal titration calorimetry

CLC-ec1 WT and K131A protein purification was carried out as described above for WT CLC-ec1 at pH 7.5 in DM, with the following differences. First, to obtain quantities CLC ec-1 needed for ITC experiments (2 mL at ~15 μM), we used cells from 2 to 3 L of pellets. Second, during the final SEC step, we eluted the protein with Cl$^-$-free buffer (buffer A: 10 mM HEPES, 150 mM Na-isethionate, 5 mM anagrade DM, pH 7.5). The protein was then dialyzed at 4 °C against the same buffer (250 mL) with buffer exchange every three hours for a total of four exchanges to remove trace Cl$^-$. Following dialysis, 2 mL of the protein sample was added to the sample chamber of the VP-ITC microcalorimeter (Malvern Panalytical, Malvern, UK). Titrant (30 mM KCl in buffer A) was injected into the sample chamber 10 μL at a time, once every four minutes for a total 180 min. Reference data were obtained by titrating buffer A into the protein-containing solution. The data were analyzed using MicroCal ITC-ORIGIN Analysis Software, with fitting using the "one set of sites" model (keeping $n = 1$).

### Reconstitution and transport assays

WT and mutant CLC-ec1 were reconstituted into liposomes[10,42]. *E. coli* polar lipid extract (Avanti Polar Lipids, Alabaster, Al, USA, 25 mg/ml) was evaporated under vacuum, washed with pentane, and then resuspended in reconstitution buffer (300 mM KCl, 40 mM citric acid, pH = 4.3 with NaOH and 21.5 mg/mL CHAPS (3-[(3-Cholamidopropyl) dimethylammonio]-1-propanesulfonate, Biotium, Freemont, CA, USA)) to a final concentration of 20 mg/ml. CLC-ec1 protein (1–2 mg/mL in DM) was then added to achieve a protein:lipid ratio of 0.4 μg/mg and a lipid concentration between 19.5 and 19.8 mg/ml. Proteoliposomes were then aliquoted into 200-μL reconstitution samples and dialyzed over two days with 4 ×1 L reconstitution buffer. Proteoliposomes were subject to four freeze-thaw cycles and then used directly for transport assays or stored at −80 °C until measured.

For flux assays[10], each 200-μL reconstitution sample was extruded through a 400-nm nucleopore filter (GE Lifesciences) to produce unilamellar proteoliposomes. Then, 60 μL of the extruded sample was buffer-exchanged using a Sephadex spin column, then diluted into 600 μL pH 4.5 flux assay buffer (300 mM K-isethionate, 2 mM citric acid and 50 μM KCl, pH 4.5). Each 200-μL reconstitution sample was used for a maximum of three flux-assay measurements (60 μL each) and the results were averaged to yield a single data point (see Source Data). For each flux-assay experimental condition, reconstitutions from at least two independent protein preparations were used. Chloride and proton transport were measured in parallel using Ag/AgCl electrodes and a micro pH electrode. Data were acquired using AxoScope 9.0 (Molecular Devices, San Jose, CA, USA) and processed using Clampfit 9.0 (Molecular Devices, San Jose, CA) and SigmaPlot 14.0 (Systat Software Inc., San Jose, CA, USA). Data and transport rates shown in the manuscript were visualized in SigmaPlot 14.0.

### Reporting summary

Further information on research design is available in the Nature Portfolio Reporting Summary linked to this article.

## Data availability

The mass spectrometry proteomics data have been deposited to the ProteomeXchange Consortium via the PRIDE [1] partner repository with the dataset identifier PXD058693 where the reference is PubMed ID: 34723319. The cryo-EM maps have been deposited in the Electron Microscopy Data Bank (EMDB) under accession codes EMD-70242 (CLC-ec1 pH 7.5); EMD-70243 (CLC-ec1 pH 4.0); EMD-70244 (CLC-ec1 pH 3.0); and EMD-70245 (CLC-ec1 K131A pH 7.5). The atomic coordinates have been deposited in the Protein Data Bank (PDB) under accession codes PDB9O95 (CLC-ec1 pH 7.5); PDB9O96 (CLC-ec1 pH 4.0); PDB9O97 (CLC-ec1 pH 3.0); and PDB9O98 (CLC-ec1 K131A pH 7.5). Structures previously published and referenced in this paper include 1OTS (CLC-ec1 crystal structure); 7RP5 (CLC-ec1 pH 4.5 cryo-EM structure); and 6V2J (CLC-ec1 QQQ mutant crystal structure) Simulation trajectories generated in this study are available at Zenodo [https://doi.org/10.5281/zenodo.17808100]. The source data underlying Figs. 2a–c, 3h, i, 4c, e, 5a, b, e–g, Supplementary Figs. 1, 2, 3a, b, 4d, 5d, 6d, 7a, c, 9, and 11d are provided as a Source Data file. Source data are provided with this paper.

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

## Acknowledgements

We thank Anindita Das and Shwetha Srinivasan for comments on the manuscript. This research was funded by NIH RO1GM113195 (M.M., R.O.D., and W.C.). For the mass spectrometry aspects of this research, financial support from MEYS/EU project OP JAK – Photomachines (CZ.02.01.01/00/22_008/0004624) is gratefully acknowledged. Access to the Centre of Molecular Structure (CMS), BioCeV, was supported by CIISB LM2023042 and ERDF "UP CIISB" (CZ.02.1.01/0.0/0.0/18_046/0015974). Cryo-EM was performed at the Stanford-SLAC Cryo-EM Center (S2C2), which is supported by the National Institutes of Health Common Fund Transformative High-Resolution Cryo-Electron Microscopy program (U24 GM129541). A.N. was supported by an administrative supplement to NIH RO1GM113195 and by a Stanford Propel Postdoctoral Fellowship.

## Author contributions

D.A. designed, performed, and analyzed the MD simulations, wrote methods; C.-T.C. collected and analyzed cryo-EM data, built and refined experimental models, modeled transport cycle, wrote methods; J.K. designed, performed, and analyzed flux-assay experiments, modeled transport cycle, wrote methods; A.R.N. prepared and purified protein samples, designed, performed, and analyzed molecular biology and ITC experiments, wrote methods; J.M.P. prepared and purified protein samples, performed and analyzed mass spectrometry experiments, wrote methods; L.F. prepared and purified protein samples, performed and analyzed mass spectrometry experiments, wrote methods; B.L.S. analyzed MD simulations; C.M. prepared and purified protein samples; P.M., R.O.D., W.C. and M.M. oversaw the research. M.M. wrote the initial draft of the paper. All authors edited and approved the manuscript.

## Competing interests

The authors declare no competing interests.
