## [Transparent Peer Review file · Nature Communications]

Molecular mechanism of exchange coupling in CLC chloride/proton antiporters

Corresponding Author: Professor Merritt Maduke

Version 0:

Reviewer comments:

Reviewer #1

(Remarks to the Author)
Please see attached review

Reviewer #2

(Remarks to the Author)
In this report by Aydin, Chien, Kreiter, Nava and colleagues, the authors investigate how CLC transporters couple the transport of Cl⁻ and H⁺. Using the well characterized E. coli CLC-ec1, the authors combine HDX, cryo-EM and molecular dynamics to investigate the conformational landscape of CLCs during transport, identifying several pH-dependent conformational rearrangements. Collectively, these approaches led to the hypotheses that inner gate dilates when the Egate moves into anion pathway and that formation of H⁺-conductive water wires are not coupled to Cl⁻ binding. These hypotheses were supported by measuring the transport rates of CLC-ec1 mutants central Cl binding site affinities. Based on these experiments, the authors propose a model for the coupled transport of Cl⁻ and H⁺. Overall, the work is intriguing and represents an important advancement in the understanding of CLC transport. However, several aspects of the work need to be clarified before it will be suitable for publication.

Comments.

1. Central to the model for the coupled transport of Cl⁻ and H⁺ is the finding that the presence of the Egate in the anion pathway leads to an opening of the inner gate. However, there are structures of cmCLC in which the Egate adopts the “down” position occupying the anion pathway and the inner gate remains closed. There are also structures of CLC-ec1 in the “middle” state, which the authors propose in Figure 6 would stabilize an open inner gate. The authors need to clarify their interpretation of the MD simulations considering the existing structural data.
2. The cryo-EM analyses of CLC-ec1 at pH 4.0 and at 3.0 revealed only a single conformation for each data set whereas a previous study identified several distinct states at pH 4.5. In the pH 3.0 processing workflow, there are several reasonable classes that were excluded from the final reconstruction. Do these correspond to alternative conformations or low-resolution states? As the HDX analysis reveals increased conformational dynamics of the transporter at pH 3.0, it would be informative to fully explore the conformational landscape of CLC-ec1. Additional structures may provide further insights into the transport mechanism of CLC-ec1.
3. Gluin, which is briefly mentioned in the introduction, is not incorporated into the transport model despite its critical role in H⁺ transport. Also, it is difficult to assess from Figure 4a, but it seems to be somewhat distant from the intracellular entrances to the water wires that were identified in the MD simulations. As the primary goal of this work is to describe coupled transport in CLCs, it is important that the authors clarify the role of Gluin in H⁺ transport.

Reviewer #3

(Remarks to the Author)
Aydin and colleagues provide a detailed study of the coupling mechanism of the prokaryotic CLC:H⁺ transporter ClCec1, using a combination of HDX-MS, cryo-EM, and MD simulations, as well as flux and transport assays, based on hypothesis

derived from the structural work. The clear added value of the work comes from the starting hypothesis that the missing steps of the mechanism could be deduced by investigating what happens at low pH (here mostly 3.0). This is a very interesting work that showcases how minute structural changes are required to understand energy coupling, and it is altogether quite elegant. The main results are that Cl⁻ binding is dependent on E-gate deprotonation, and that coupling does not depend on Cl⁻ binding. This, combined with the vast amount of literature available, allows the authors to propose an updated transport cycle.

I do, however, have one major concern: the experiments performed at pH 3 by HDX-MS and their interpretation (see below). Besides this concern, which I hope the authors will be able to address, I believe this article to be intellectually stimulating and well conceived. I honestly think that most of their work is still convincing without that HDXMS dataset.

The comparison of HDX values at different pH levels is problematic when it is so close to the pH values of quenching. I would be very cautious with the interpretation. The authors applied a correction of actual H/D exchange by increasing the incubation time, but this only works when H/D exchange follows a first-order reaction. This linear dependency between pH and rate of exchange has been convincingly demonstrated for pH values above 4 (refs: Chapters 1 and 2 of the book "Hydrogen Exchange Mass Spectrometry of Proteins Fundamentals, Methods, and Applications Edited by David D. Weis; and the seminal article from Englander in 1993 Bai Y, Milne JS, Mayne L, Englander SW. Primary structure effects on peptide group hydrogen exchange. *Proteins*. 1993 Sep;17(1):75-86. doi: 10.1002/prot.340170110. PMID: 8234246; PMCID: PMC3438223.). However, below that threshold, the contribution of acid-catalysed HD exchange becomes significant, and the process deviates from a simple linear relationship. In such cases, time correction is likely inaccurate, and what we observe is a convolution of effects from both the increased incubation time and the change in pH. Consequently, the observed differences cannot be attributed solely to pH changes. The fact that the protein as a whole shows substantial exchange while remaining, on average, well folded supports the notion that we are likely observing a mixture of effects. Accurately correcting for this, especially in a complex system such as a mixed protein-detergent micelle (a scenario not even considered when the fundamental equations of H/D exchange were derived), is highly challenging. For me, this represents a major issue. For this reason, the following statements are problematic:

Line 67: "When we lowered the pH from 4.0 to 3.0, we observed a sharp increase in protein conformational dynamics detected by hydrogen deuterium exchange spectrometry". It might be true, but it is likely that this increase at pH 3 is not caused by increased dynamics. Another interpretation is that the exchange reaction at pH 3.0 is not a ten fold faster than at pH 4.0, and maybe only seven time faster, for example, as the linear relationship is not valid anymore. As a result, the increase in deuteration time does not match the increase in the exchange rate, leading to an increased exchange that is mostly due to the longer deuteration time.

line 83 "with virtually no limitation in experimental conditions (pH, temperature, buffer composition)". The pH at which you can work is a limitation, as explained above.

Line 105: "Fig.2b shows a differential heat map for deuterium incorporation observed in CLC-ec1 at pH 3.0 compared to pH 4.0. In this case, many peptides displayed increased HDX at the lower pH. This result indicates a second conformational transition, encompassing a broad swath of the protein." "With the current knowledge of HDX chemistry in proteins, I cannot say that this claim is supported by the data. It might be true, but it might just as well be an artifact. Therefore, the section describing a second transition needs to be severely revised or removed.

Line 121: "Peptides 146-163 (helix F, orange) and 110-116 (helix D, teal green) (plots at right) represent the second pH-dependent conformational transition, detected by the shift in the deuterium uptake plot occurring at pH 3.0 compared to 3.5." same as above.

Line 273: "Our HDX-MS results align with those previous results and reveal additional conformational dynamics that occur at lower pH values, down to pH 3.0." same as above.

Minor comments

Line 4: osteopros

Line 139: fig c e should be figure b e

Fig 2b: I'm confused by the axis line in the uptake plots. The rate of exchange is faster at high pH than at low pH. It is ~100-fold faster at pH 6.5, so, in order to do valid comparisons, the sample is incubated 100x longer at low pH 4.5 than at high pH 6.5. In the end, the uptake values at, for example, 2000 s at pH 4.5 are compared with 20 s at pH 6.5, which makes sense. Then why are the time points corresponding to pH 6.5 longer than those taken at lower pH values? Isn't it supposed to be the other way around? I guess it comes down to the normalization, but then what exactly do you mean by 'exchange times are normalized to pH 3.0'?

Line 203 "suggesting" – not indicating. MD are predictions.

Fig 3.j. I am not a cryo-EM expert, but is a resolution above 3.2 Å, as reported for the pH 3 and pH 4 structures, sufficient to confidently validate the change in orientation of F357 between these structures?

Line 258: "K131 mutations had no discernable effect on E148A Cl⁻ transport rates (Fig. 5g, Table 2). These results align with the "stalled transport cycle" model for K131 mutants: when these low Cl⁻-binding mutant transporters are untethered from obligate Cl⁻/H⁺ coupled transport, Cl⁻ movement becomes faster, unimpeded by the need to wait for a stalled proton-transport step." Can the authors develop this hypothesis? The point made in this section is unclear to me.

Fig.5: What does z score of 11.1 vs 7.9 mean, for a non-expert?

Line 236: "Importantly, we structured this flux-assay experiment to ensure that any H⁺ movement into the vesicles was a result of Cl⁻-dependent H⁺ pumping rather than leakage". How did you do that? Can you be more specific? Edit: I found the explanation in the figure legend, but it would be nice to have a clearer pointer to it in the main text.

Version 1:

Reviewer comments:

Reviewer #1

(Remarks to the Author)
Please see attached review

Reviewer #2

(Remarks to the Author)
The manuscript has been suitably improved and is ready for publication.

Reviewer #3

(Remarks to the Author)
The authors have fully addressed my concerns. The manuscript is now much clearer, and I appreciate the effort they invested in improving the clarity of the text. What might feel like “explaining the obvious” is very helpful for guiding the reader through the concepts and strengthening the accessibility of the paper. These revisions greatly enhance the readability and - as a corollary - impact of the work.

We thank the reviewers for their time in providing positive feedback and thoughtful suggestions on our manuscript. We address each comment below and have provided a revised manuscript in which changed regions of text are highlighted. We believe the revisions have improved the clarity and overall presentation of the work. We appreciate the reviewers' role in helping us strengthen the manuscript.

Reviewer 1

1) The physiological relevance of the pH 3.0 structure is unclear for a few reasons:

- a. In extreme acid response, only the outside of the bacteria is at such low pH. Presumably the inside remains close to pH5 or the bacteria die. This should be discussed and the implications for the findings considered.
- b. The HDX-MS suggests that large regions of the protein are water accessible at pH3.0, including TM regions that should not be water accessible. How is this physiologically possible? Is it possible the protein is only partially stable and the cryo structures are coming from a subset of the population? This should be discussed in the paper.
- c. Given the significant change in HD exchange between pH 3 and 4.5, electrophysiology at pH 3.0 is needed. Is the stoichiometry still 2:1? How does current change?

Thank you for requesting clarification on these important points.

To address points (a) and (c), we have added the following text to the revised manuscript (lines 64-69):
"Motivated by CLC-ec1's physiological function of mediating extreme acid tolerance within the mammalian stomach ⁴, we explored its pH-dependent conformational changes at even lower pH. Although only the extracellular face encounters acidity *in vivo*, symmetric pH conditions greatly simplify biophysical assays. Crucially, electrophysiological studies confirm that at symmetric pH 3 CLC-ec1 remains fully active, preserving a 2:1 Cl⁻/H⁺ coupling ratio ¹ and exhibiting increased transport rates as the pH is lowered symmetrically from 4.5 to 3.0 ³⁵."

Regarding point (b), we agree that the high deuteration levels observed at very low pH may raise questions about protein stability. Accordingly, we evaluated multiple indicators supporting the protein's structural integrity and relevance of the data to protein conformational change rather than instability. Our initial consideration is that exchange data must be evaluated in the context of the kinetics. What we observe is a progressive increase in exchange at the intra- and extracellular surfaces and along the inner ion pathway, with these changes propagating outward to surrounding regions. This pattern is consistent with inside-out propagation of local dynamics, rather than a global loss of stability. Second, we note that although the exchange is extensive, the uptake remains region-specific: the deepest hydrophobic core regions are still protected, consistent with localized dynamic changes rather than global unfolding. We also note that precipitation or aggregation would be expected to generate non- or slowly exchanging populations, which was not observed. Third, to test the CLC-ec1 stability, we analyzed CLC-ec1 by size-exclusion chromatography (SEC) after incubation at pH 3.0 for several time periods. The elution profiles (Supplementary Fig. 2) showed a homogeneous population, with no evidence of aggregation or degradation. Fourth, we incorporated a control directly into the HDX-MS workflow: CLC-ec1 was pre-incubated for 63000s (the longest HDX time point) at the respective pH and then subjected to a short D2O pulse corresponding to the regular 20s labeling time. When overlaid with the standard 20-s data (resulting in a hexaplicate measurement), the uptake matched closely. Any global unfolding or degradation would have produced increased deuterium incorporation in this control. Thus, within the experimental timescale, the overall CLC-ec1 population remains stable and homogeneous. These points have been incorporated into our revised description of the HDX results (lines 90-139). For the cryo-EM analysis at pH 3.0, the sample displayed a monodisperse peak on SEC with no evidence of aggregation. This supports the supposition that the pH 3.0 cryo-EM structure reflects the broader population and not merely a stable subset.

2) A more direct comparison between a recently published pH 4.5 structure (Fortea et al.) and the presented structures would be an important addition to understand what is truly new here. Aside from the partially obtained twist structure in that paper, how are the obtained structures different or similar to the full turn structure? How do the regions considered to be important differ?

We appreciate the reviewer's suggestion to directly compare our structures with the pH 4.5 "Turn" structure reported by Fortea et al. (PDB: 7RP5). We calculated RMSDs between 7RP5 and our pH 3 and pH 4 structures and have added this information to the revised manuscript text (lines 156-177), additions highlighted below, and in a new Supplementary Figure:

"Overall, our pH 4.0 structure closely resembles 7RP5, with large RMSDs occurring mostly in loop regions that show low Q-scores in our maps (**Supplementary Figure 7a**). The low Q-scores reflect the intrinsic flexibility of these loop regions and contribute to greater uncertainty in model-to-model comparisons in these regions. The most notable difference between our pH 4.0 structure and 7RP5 is the orientation of helix A, which angles away from the membrane in our structure compared with 7RP5 (**Supplementary Figure 7b**). We hypothesized that this difference stems from the membrane mimicking lipid nanodiscs used in our study versus detergent DM in 7RP5. Supporting this hypothesis, our pH 4.0 structure in DM reveals a helix-A orientation like that in 7RP5 (**Supplementary Figure 7b**). This orientation change is likely driven by the stark physicochemical differences between the nanodisc's flat lipid bilayer and the small, curved detergent micelle, causing the helix to adjust its tilt angle to optimize its hydrophobic interactions. Future work should therefore investigate the functional role of specific lipid headgroups and their interactions with helix A

In the pH 3.0 cryo-EM structure, there is a subtle conformational change compared to pH 4.0, with an overall C- α r.m.s.d of 0.92 Å. Notably, the pH 3.0 conformation resembles the "QQQ" CLC-ec1 crystal structure (6V2J) (overall C- α r.m.s.d 0.84 Å) more closely than the pH 4.0 structure (overall C- α r.m.s.d 1.27 Å) (**Fig. 3h**). Compared with the pH 4.0 structure (**Fig. 3h**) and with the pH 4.5 structure 7RP5 (**Supplementary Figure 7c**), the pH 3.0 map indicates a more dynamic conformational landscape: the H-I loop shows high flexibility, as reflected by low Q scores, and Helix A is absent. These cryo-EM features agree with our HDX results and with the multiple heterogeneous particle classes seen in the pH 3 dataset (**Supplementary Fig. 6a**). Although many regions are flexible at pH 3.0, the conserved pore region remains well resolve."

Supplementary Figure 7: Comparison to pH 4.5 structure, PDBID: 7RP5. (a) RMSD between 7RP5 and the pH 4.0 structure presented here. The large changes are predominantly in loop regions where Q-scores are low and therefore model-to-model comparisons have low certainty (b) Structure overlay showing change in orientation of helix A in the pH 4.0 structure compared to pH 7.5 and to pH 4.5 (7RP5). Our additional structure at pH 4.0 in DM shows that the orientation of helix A depends on the membrane mimetic used. (c) RMSD between the pH 3.0 structure and 7RP5 (pH 4.5).

3) The proposed uncoupling shows E_{gate} moving from the "middle" to the "up" position, allowing Cl⁻ to flow out without protonation. What data supports this notion? Previous work has demonstrated that this should be prohibitively unfavorable. So, why would the stalled proton transport state not be the protonated form of E_{gate}—especially at these low pH values?

Thank you for raising this point. We acknowledge that the uncoupling model for the K131 mutants shown in Figure 6b, which depicted E_{gate} in the "up" position as unprotonated, lacked sufficient justification. We therefore removed this panel and revised the Discussion, focusing on the K131 mutants' most striking phenotype (a 20-fold reduction in transport rates) and offering a plausible explanation for the slight Cl⁻/H⁺ uncoupling observed, lines 400-418:

"K131 is universally conserved across the CLC family, including both transporter and channel homologs. This conservation, combined with K131's location – entirely buried within the transmembrane domain, which is rare for lysine residues⁴⁴ – suggests that it plays a crucial functional role. Indeed, mutations at this position in CLC channels shift voltage dependence and lead to myotonia^{45,46,47}. In CLC-ec1, K131 mutations weaken Cl⁻ binding affinity and reduce Cl⁻/H⁺ transport rates (**Fig. 5, Table 2**). Results from experiments on E_{gate} mutant transporters (**Fig. 5g**) indicate that the transport slowdown stems from inhibition at one or more of the E_{gate}-dependent steps. We propose that key bottlenecks occur during transitions where Cl⁻ and E_{gate} compete for binding to the S_{ext} and S_{cen} sites. Although K131 mutations likely reduce E_{gate}'s binding affinity at these sites, its physical tethering confers a competitive advantage over free Cl⁻ ions. In the context of the exchange mechanism (**Fig. 6**), transitions D \rightleftharpoons E, D \rightarrow C, and E \rightarrow A will be slow, stalling the transport cycle. This model for stalled transport is consistent with the K131A structure, which shows that the cryo-EM density for E_{gate} is less affected compared to that of Cl⁻. The model also aligns with the largely preserved coupling

seen in K131 mutants, where mild uncoupling raises the Cl^-/H^+ stoichiometry from 2 to ~3. A more severe loss of E_{gate} affinity would allow freer Cl^- movement and would drive Cl^-/H^+ stoichiometry higher, as seen in other mutants with low Cl^- binding affinity¹². The mild uncoupling in K131 mutants could result from loss of hydrogen bonds between K131 and inner-gate loop residues A104 and G106, slightly destabilizing the inner gate and allowing an extra Cl^- ion to slip through roughly once per transport cycle (**Supplementary Fig. 12**).”

4) The K131 results are interesting. It is argued that the impacts of this mutant are due to the decreased affinity of Cl^- to the central binding site. But the stability of E_{gate} in this site would also be altered. Since there is no change in the rates for E148A, the rate of release from the central site is not limiting in the absence of proton coupling. Yet the rate of transport when coupling is present is slowed. So, this mutant stalls a proton-coupled step and its impact on the stability of E_{gate} would be a more logical explanation.

As discussed in our response to point 3 (above), we have rewritten the discussion section concerning the impact of the mutations. We also revised the results section describing the E148A mutant flux assays to more accurately describe how E_{gate} mutation data can be interpreted, lines 286-288:

“If K131 mutations slow transport as a direct result of weakened Cl^- binding, then they should affect Cl^- transport rates in both WT and E148A backgrounds. On the other hand, if K131A mutations slow transport because they stall one of the E_{gate} steps in the coupled transport cycle (either protonation/deprotonation of E_{gate} or $E_{\text{gate}}/\text{Cl}^-$ competition), then they should have no effect in the E148A background.”

5) The simulations show that once E_{gate} is deprotonated in the down_out conformation (i.e., deprotonation of state B in Fig 6) it rapidly moves to the central Cl^- site and pushes two Cl^- out. This is not consistent with the cycle shown in Fig 6 which instead shows multiple metastable intermediates D and E, which makes the latter hard to interpret and connect with the simulation results. What justifies these intermediates?

We have added a paragraph to the discussion to explain the rationale for the intermediates, lines 373-386:

“Our simulations identified two key elements that shape the proposed transport mechanism: inner-gate opening and a new water wire. We model inner-gate open states as transient states because inner-gate opening appears in simulation but has not been captured in low-energy structural snapshots. In the simulations described here, 2 Cl^- ions are expelled to the extracellular solution, leaving a pathway with no bound Cl^- . Structural data show that single- Cl^- occupancy states are stable^{7, 43}, so we model them as states D and E. To represent the short-lived inner-gate openings seen in simulations, we also include transient, inner-gate-open variants of these states (dashed boxes) that would permit Cl^- movement to and from the intracellular solution. Although the simulation did not discretely capture these single-occupancy states, this is reasonably explained by the challenges of observing Cl^- entry in simulations. Ions already in the permeation pathway can easily overcome small energy barriers to dissociate from the binding site within a simulation timescale, while the diffusion of Cl^- to the binding site is a longer timescale process not captured within the length of our simulations. The primary takeaway is that the simulations demonstrate the feasibility of the inner-gate open state and thus support its role in the transport mechanism.”

6) Given that the transition from B to D (with an additional Cl^- release to the outside) is almost instantaneous, this step must be downhill and nearly barrierless. What would cause the system to reverse? I.e., how would it go uphill from B to A?

We ran two simulations from different starting points: one matching State B of our model and a second identical to State B but with E_{gate} deprotonated. The B to D transition is rapid (occurring within ~50-100 ns) under the second condition but is not observed under the first condition. The B to A transition was not observed in either simulation and would likely require longer simulation times or different conditions to observe.

In the revised manuscript we expanded the Figure 6 legend to state that the relative rates of the mechanism’s steps depend on ion concentrations, ion gradients, and the transmembrane voltage.

7) If it really is reversible then simulations starting in A should also push Cl^- in. What happens when simulations start in A? Was B to A ever observed? I.e., does protonated E148 ever relocate to above the bound Cl^- such that deprotonation could push Cl^- in.

We have not run any simulations starting in A, so we can't speak to the MD timescale or conditions required for observing the A to B or B to E transitions in simulation. That said, if we did run a simulation starting in A, we expect we would not see a transition from B to A because of the limited time scales we can sample via simulation.

8) How would the transmembrane voltage influence the proposed steps in Fig 6? Was this factor included in the simulations?

We have not studied the effects of transmembrane voltage in this study. We have edited the text to note that transmembrane voltage is expected to influence the steps (Figure-6 legend) and to make it clear that our simulations were not performed under a transmembrane voltage (lines 618-619)

9) What are the residues with a very large RMSD relative to the QQQ structure and why are they so different?

The main region with a large RMSD relative to the QQQ structure is the H-I loop. In our pH 3 structure, this region is poorly resolved, as indicated by the low Q-scores in Supplementary Figure 6d. In the pH 4 structure, Q207 on the H-I loop interacts with E117. At pH 3, protonation of E117 likely disrupts this interaction, leading to destabilization and increased flexibility of the H-I loop. We have added this information to the Figure-3 legend.

10) The reported stoichiometry at pH 4.5 (2.1 ± 0.1 , Table 2) aligns with the classical value of 2.2, yet the authors consistently refer to it as 2. Are the authors suggesting an exact 2:1 exchange ratio? Clarification is important, as a non-integer ratio indicates multiple co-existing exchange pathways (Not cited but should be: Mayes et al., *J. Am. Chem. Soc.* 2018, 140, 1793–1804). Or is there a detail I may have overlooked?

Given the experimental uncertainty, the measured stoichiometry is indistinguishable from the integer value 2, and therefore it cannot be concluded that the stoichiometry is non-integer. That said, we recognize this is an interesting and important point that warrants discussion, which we have added to the revised manuscript, lines 424-429:

“An alternative to our proposed 2:1 mechanism was previously suggested through elegant kinetic modeling, showing that multiple distinct pathways can collectively yield the experimentally observed stoichiometry⁴⁹. Importantly, that modeling used structures of an inactive CLC-ec1 conformation and predated recognition that E_{gate} can adopt an “out” conformation; updating the modeling could clarify how these alternate pathway ensembles might complement the mechanism we propose.”

11) Figure 1 links extracellular pore conformation to E_{gate} rotamers, i.e., “open” with E_{gate} “out”, and “closed” or “occluded” with other E_{gate} rotamers. But in the proposed mechanism (Figure 6), all E_{gate} rotamers appear associated with an “open” extracellular pore. Are the authors proposing that the extracellular gate remains open throughout the cycle? This apparent inconsistency needs clarification.

Thank you for pointing this out. The depiction of all E_{gate} rotamers being associated with the “open” extracellular pore was an oversight on our part. We have updated Figure 6 so that each cartoon reflects the outer-pore dimensions observed in the relevant structures.

12) The mechanism behind inner gate opening below pH 4 is not addressed. Is this linked to protonation of residues such as E203? Prior computational studies indicate protonation of E113 and E203 at pH 3 disrupts ion pairs with R28' from the twin monomer (Yue et al., *Biophys. J.* 2023, 122, 1068–1085). This should trigger gate opening and water entry. The absence of helix A in the QQQ structure may further support this. Do the authors have evidence supporting or refuting a role for E113, E202, or E203 protonation in this process? If so, where do the protons go once the gate opens?

We did not study the detailed mechanism of the inner-gate opening that we observed in the pH 3 simulation. Similarly, we did not investigate the mechanism of the low-pH conformational change that moves helix A out of the way of the proton-permeation pathway (we think this is the “inner gate” being referred with respect to Yue et al.). We did not study the effects of protonation on E113, E202, or E203.

We believe these points are not essential to the main questions addressed in the manuscript.

13) The manuscript frequently refers to “water wires” and quantifies their presence (Figure 4e), yet I could not locate a clear definition. Please define what is meant by a “water wire” in this context. I note that it has been pointed out a number of times in the literature that a water wire without an excess proton in it is quite different from simply a water wire (see Li et al., *Proc. Natl. Aca. Sci.* 2021, 118, e2113141118). This should be discussed and cited.

We have defined water wires as chains of hydrogen-bonded water molecules (lines 31 and 184). We have updated the methods to clarify that we did not model an excess proton when studying water wire formation and have revised the text to discuss this point and the point that we did not simulate explicit proton transport (see response to point 15, below)

14) MD simulations of the pH 3 structure identified three water wires (Figure 4a). The authors highlight that water wire 1 runs through the Cl⁻ permeation pathway and suggest its involvement in the counterclockwise transport cycle (Figure 6). Under this scenario, do protons supposedly permeate through the water wire 1? That is, do Cl⁻ and H⁺ share the same permeation pathway?

We have updated the text to make this point clearly, lines 394-397:

“In this conformation, water-wire 1 is connected to E_{gate}, but protonation of E_{gate} is not yet energetically favorable, with only a single Cl⁻ ion present in the pathway. Upon entry of a second Cl⁻ (state **C**), protonation of E_{gate} becomes energetically favorable, enabling proton transfer via water-wire 1.”

15) While water wires (without excess protons in them) are often used to infer or guess proton transport (PT) behavior, their presence alone does not guarantee PT. For example, aquaporins form water wires but block proton flow. Proton conduction involves the formation of transient hydrogen-bonded chains that excess protons themselves help assemble, as shown in CIC-ec1 (Lee et al., *Biophys. J.* 2016, 110, 1334–1345). I recommend the authors consult relevant literature (Peng et al., *J. Phys. Chem. B* 2015, 119, 9212–9218; Li et al., *Proc. Natl. Aca. Sci.* 2021, 118, e2113141118) and references therein. They should also point out that they are not simulating explicit proton transport with their MD.

We have added a paragraph address this important point (lines 430-440)

“Our model posits that proton transport occurs along connected water wires identified in our MD simulations. These contiguous hydrogen-bonded chains provide the structural and electrostatic continuity required for Grothuss-like hopping through a hydrophobic protein, marking the minimal requirement for proton transport. In our simulations, we did not model an excess proton in the water wires, and we did not simulate explicit proton transport; these simplifications allowed us to focus computational effort on sampling structural dynamics and hydration patterns. In simulations of the CLC-ec1 high-pH conformation, inclusion of an excess proton was found to draw additional solvated water into the protein core and create broader water-wire structures compared to simulations without the excess proton⁵⁰. Explicit proton-inclusive simulations^{40, 50, 51} would be an appropriate next step to quantify proton hopping kinetics and free-energy barriers, building on the structural and functional insights established here.”

In summary, while CIC-ec1 is generally thought to operate without major conformational changes within the typical pH range of 4–7.5, experimental evidence here suggests altered behavior below pH 4. This study may provide insights into the structure and function of CIC-ec1 under more acidic conditions. However, the PT mechanism at pH 3 remains unclear. I encourage the authors to not only address the issues listed above but to also expand their discussion on this issue and suggest future directions to address the remaining open questions.

Thank you for the constructive comments. The manuscript is improved and more clearly argued as a result of the revisions.

Reviewer 2

In this report by Aydin, Chien, Kreiter, Nava and colleagues, the authors investigate how CLC transporters couple the transport of Cl⁻ and H⁺. Using the well characterized *E. coli* CLC-ec1, the authors combine HDX, cryo-EM

and molecular dynamics to investigate the conformational landscape of CLCs during transport, identifying several pH-dependent conformational rearrangements. Collectively, these approaches led to the hypotheses that inner gate dilates when the Egate moves into anion pathway and that formation of H⁺-conductive water wires are not coupled to Cl⁻ binding. These hypotheses were supported by measuring the transport rates of CLC-ec1 mutants central Cl binding site affinities. Based on these experiments, the authors propose a model for the coupled transport of Cl⁻ and H⁺. Overall, the work is intriguing and represents an important advancement in the understanding of CLC transport. However, several aspects of the work need to be clarified before it will be suitable for publication.

Comments.

1. Central to the model for the coupled transport of Cl⁻ and H⁺ is the finding that the presence of the Egate in the anion pathway leads to an opening of the inner gate. However, there are structures of cmCLC in which the Egate adopts the “down” position occupying the anion pathway and the inner gate remains closed. There are also structures of CLC-ec1 in the “middle” state, which the authors propose in Figure 6 would stabilize an open inner gate. The authors need to clarify their interpretation of the MD simulations considering the existing structural data.

We have clarified our interpretation of the MD simulations as revealing a transient rather than a stable inner-gate open, adding a clarifying sentence in the Results section as well as a paragraph to the revised manuscript.

Lines 331-333: “Since this conformation has not been seen in any static structures, including structures in the absence of Cl²⁶, we interpret that the simulations have captured a transient, functionally important state

Lines 373-386, “Our simulations identified...” (See response to Rev 1 point #5),

2. The cryo-EM analyses of CLC-ec1 at pH 4.0 and at 3.0 revealed only a single conformation for each data set whereas a previous study identified several distinct states at pH 4.5. In the pH 3.0 processing workflow, there are several reasonable classes that were excluded from the final reconstruction. Do these correspond to alternative conformations or low-resolution states? As the HDX analysis reveals increased conformational dynamics of the transporter at pH 3.0, it would be informative to fully explore the conformational landscape of CLC-ec1. Additional structures may provide further insights into the transport mechanism of CLC-ec1.

The reviewer is correct that approximately 160,000 particles were discarded during 3D classification in Relion. These particles could be reconstructed to comparable nominal resolution; however, the resulting maps showed incomplete density in several regions of the protein. Specifically, density was missing for the I–J loop, the B–C loop, helix R, and helix D. To enable building a complete atomic model, we therefore focused on the subset of 80,122 particles that yielded the most complete map. The absence of density in the excluded classes reflects increased flexibility in these regions, consistent with our HDX results that conformational dynamics are enhanced at pH 3.0. We have added this information to the Figure legend:

“Supplementary Figure 6: Cryo EM workflow and validation data for CLC-ec1 at pH 3.0. (a) Cryo-EM data processing workflow. The ~160,000 particles that were discarded during 3D classification in Relion could be reconstructed to comparable nominal resolution as our final model; however, the resulting maps showed missing or incomplete density in several regions of the protein. Specifically, density was missing for the I–J loop, the B–C loop, and helix D. To enable building a complete atomic model, we focused on the subset of 80,122 particles that yielded the most complete map. The absence of density in the excluded classes reflects increased flexibility in these regions, consistent with our HDX results that conformational dynamics are enhanced at pH 3.0.”

3. Gluin, which is briefly mentioned in the introduction, is not incorporated into the transport model despite its critical role in H⁺ transport. Also, it is difficult to assess from Figure 4a, but it seems to be somewhat distant from the intracellular entrances to the water wires that were identified in the MD simulations. As the primary goal of this work is to describe coupled transport in CLCs, it is important that the authors clarify the role of Gluin in H⁺ transport.

Thank you for highlighting this point - we recognize it was confusing to have Glu_{in} highlighted in Figure 1 and then not included in the final model. We have updated the Fig-1 legend to clarify that Glu_{in} is not strictly required:

Figure 1: CLC structure overview. (a) Side view of CLC-ec1 (pdb 1OTS). The Cl⁻ and H⁺ permeation pathways are indicated by green and blue arrows, respectively. Bound Cl⁻ is shown as a green sphere. Key glutamate residues E_{gate} and Glu_{in} are shown space filled. E_{gate} physically gates the Cl⁻ permeation pathway and serves as an essential conduit for H⁺; Glu_{in}, though not strictly required, supports fast H⁺ transport along the intracellular portion of the H⁺ permeation pathway^{10, 73}.

In addition, we explained why we excluded Glu_{in} from Fig 4b and we added Supplementary Fig. 8, showing examples of water wires bordered by Glu_{in} and those that are not:

Lines 200-204: “Water wires 2 and 3 offer a plausible explanation for how a proton moving to the inside is coupled to two chloride ions moving out, as illustrated in **Fig. 4b**. Glu_{in} (**Fig. 1a**) is not depicted in this diagram because, although it substantially enhances H⁺ transport rates¹⁷, it is not strictly required for H⁺ transport¹⁰. **Supplementary Fig. 8** shows examples of water wires that are bordered by Glu_{in} as well as those that are not.”

Finally, we added an explanation for leaving Glu_{in} out of the final transport model:

Lines 363-365: “Glu_{in} (**Fig. 1a**) is not included in the model because it is nonessential in CLC-ec1¹⁰ and is absent in some CLC transporters⁴³, indicating it is not required for the mechanism.”

Reviewer #3 (Remarks to the Author):

Aydin and colleagues provide a detailed study of the coupling mechanism of the prokaryotic CLC:H⁺ transporter ClCec1, using a combination of HDX-MS, cryo-EM, and MD simulations, as well as flux and transport assays, based on hypothesis derived from the structural work. The clear added value of the work comes from the starting hypothesis that the missing steps of the mechanism could be deduced by investigating what happens at low pH (here mostly 3.0). This is a very interesting work that showcases how minute structural changes are required to understand energy coupling, and it is altogether quite elegant. The main results are that Cl⁻ binding is dependent on E-gate deprotonation, and that coupling does not depend on Cl⁻ binding. This, combined with the vast amount of literature available, allows the authors to propose an updated transport cycle.

I do, however, have one major concern: the experiments performed at pH 3 by HDX-MS and their interpretation (see below). Besides this concern, which I hope the authors will be able to address, I believe this article to be intellectually stimulating and well conceived. I honestly think that most of their work is still convincing without that HDXMS dataset.

The comparison of HDX values at different pH levels is problematic when it is so close to the pH values of quenching. I would be very cautious with the interpretation. The authors applied a correction of actual H/D exchange by increasing the incubation time, but this only works when H/D exchange follows a first-order reaction. This linear dependency between pH and rate of exchange has been convincingly demonstrated for pH values above 4 (refs: Chapters 1 and 2 of the book “Hydrogen Exchange Mass Spectrometry of Proteins Fundamentals, Methods, and Applications Edited by David D. Weis; and the seminal article from Englander in 1993 Bai Y, Milne JS, Mayne L, Englander SW. Primary structure effects on peptide group hydrogen exchange. *Proteins*. 1993 Sep;17(1):75-86. doi: 10.1002/prot.340170110. PMID: 8234246; PMCID: PMC3438223.). However, below that threshold, the contribution of acid-catalysed HD exchange becomes significant, and the process deviates from a simple linear relationship. In such cases, time correction is likely inaccurate, and what we observe is a convolution of effects from both the increased incubation time and the change in pH. Consequently, the observed differences cannot be attributed solely to pH changes. The fact that the protein as a whole shows substantial exchange while remaining, on average, well folded supports the notion that we are likely observing a mixture of effects. Accurately correcting for this, especially in a complex system such as a mixed protein–detergent micelle (a scenario not even considered when the fundamental equations of H/D exchange were derived), is highly challenging. For me, this represents a major

issue. For this reason, the following statements are problematic:

We thank the reviewer for this insightful comment and fully agree that exchange-rate correction at low pH must be handled with caution and below pH 4.0 it was not done correctly. First, we would like to clarify the experimental scheme applied in this study. We did not adjust the incubation times during labeling, meaning that at lower pH the samples were not incubated for progressively longer periods. Instead, all samples across the different pH conditions were collected at exactly the same time points, and correction was applied only post-acquisition by multiplying the time scale by the appropriate rate factors. The sampling scheme, therefore, was chosen to provide time points that align across conditions after correction, at least in the upper (4.5-6.5) pH region. As the reviewer correctly notes, the simplistic linear correction is valid only at higher pH values, while at lower pH the contribution of acid-catalyzed exchange leads to deviations from linearity. To address this limitation, we revised the text and figures. For the pH 6.5 to 4.5 transition, we retain the post-corrected differential plots, which highlight the structural effects of protonation. For lower pH values, we rely on uncorrected data instead, as the observed changes are unambiguous. Lowering the pH below 4.0 resulted in markedly increased dynamics, substantial reorganization of hydrogen bonding, and likely changes in hydration, which is reflected in faster HDX kinetics and higher deuteration levels across many regions of CLC-ec1, even though theory would predict slower kinetics at lower pH.

Because this issue is of broader relevance, we additionally provide a complementary set of uptake plots post-corrected using rate factors calculated with the SPHERE server, which is based on the exchange constants cited by the reviewer. While we recognize that these constants were derived using artificial sequences and under conditions different from those in membrane protein studies, they nonetheless provide a useful approximation. This view is further supported by recent work (Moroco et al., High-Throughput Determination of Exchange Rates of Unmodified and PTM-Containing Peptides Using HX-MS), where the authors reported good agreement between predicted exchange rates and experimental measurements. We hope that such data interpretation is acceptable.

To address the specific issues raised below, we rewrote the HDX section of the manuscript (lines 89-139) to focus on interpretation without correction at the low pH values (as described above). In addition, we rewrote the methods section (lines 501-521). We also clarified that deuteration times were the same for all samples (lines 510-511 - "Aliquots were collected at 20s, 63s, 200s, 633s, 2000s, 6325s, 20000s, 63000s under all pH conditions").

Line 67: "When we lowered the pH from 4.0 to 3.0, we observed a sharp increase in protein conformational dynamics detected by hydrogen deuterium exchange spectrometry". It might be true, but it is likely that this increase at pH 3 is not caused by increased dynamics. Another interpretation is that the exchange reaction at pH 3.0 is not a ten fold faster than at pH 4.0, and maybe only seven times faster, for example, as the linear relationship is not valid anymore. As a result, the increase in deuteration time does not match the increase in the exchange rate, leading to an increased exchange that is mostly due to the longer deuteration time.

See green-highlighted text above

line 83 "with virtually no limitation in experimental conditions (pH, temperature, buffer composition)". The pH at which you can work IS a limitation, as explained above.

We eliminated this text.

Line 105: "Fig.2b shows a differential heat map for deuterium incorporation observed in CLC-ec1 at pH 3.0 compared to pH 4.0. In this case, many peptides displayed increased HDX at the lower pH. This result indicates a second conformational transition, encompassing a broad swath of the protein. "With the current knowledge of HDX chemistry in proteins, I cannot say that this claim is supported by the data. It might be true, but it might just as well be an artifact. Therefore, the section describing a second transition needs to be severely revised or removed.

See green-highlighted text above

Line 121: “Peptides 146-163 (helix F, orange) and 110-116 (helix D, teal green) (plots at right) represent the second pH-dependent conformational transition, detected by the shift in the deuterium uptake plot occurring at pH 3.0 compared to 3.5.”.

See green-highlighted text above

Line 273 : “Our HDX-MS results align with those previous results and reveal additional conformational dynamics that occur at lower pH values, down to pH 3.0.”

See green-highlighted text above

Minor comments

Line 4: osteoprosis

Corrected to “osteopetrosis”

Line 139: fig c e should be figure b e

corrected

Fig 2b: I’m confused by the axis line in the uptake plots. The rate of exchange is faster at high pH than at low pH. It is ~100-fold faster at pH 6.5, so, in order to do valid comparisons, the sample is incubated 100× longer at low pH 4.5 than at high pH 6.5. In the end, the uptake values at, for example, 2000 s at pH 4.5 are compared with 20 s at pH 6.5, which makes sense. Then why are the time points corresponding to pH 6.5 longer than those taken at lower pH values? Isn’t it supposed to be the other way around? I guess it comes down to the normalization, but then what exactly do you mean by ‘exchange times are normalized to pH 3.0’?

We appreciate the reviewer’s thoughtful comment. The described approach, adjusting the labeling times so that, for example, 20s at pH 6.5 corresponds to 2000s at pH 4.5, is one way to perform cross-pH comparisons. However, this strategy becomes impractical across a broad pH range, as it would require either extremely short sampling or excessively long incubations. As described above, we employed an alternative approach: collecting data at identical time intervals for all pH conditions and then applying post-correction factors to align the data on a comparable time scale. Normalization to a selected reference pH is then achieved by multiplying the higher-pH time scales by the calculated rate-correction factors. To minimize confusion, we now present the uncorrected uptake plots in the main figures, while providing the post-corrected data set (using SPHERE-derived theoretical correction factors) as supplementary material.

Line 203 “suggesting” – not indicating. MD are predictions.

Suggested change has been made (line 233)

Fig 3.j. I am not a cryo-EM expert, but is a resolution above 3.2 Å, as reported for the pH 3 and pH 4 structures, sufficient to confidently validate the change in orientation of F357 between these structures?

At a resolution of ~3.2 Å, the protein backbone and most bulky side chains, including aromatic residues such as F357, are well resolved in cryo-EM density maps. Although individual hydrogen atoms cannot be visualized, the orientation of aromatic rings can generally be assigned with confidence at this resolution. In our maps, the density corresponding to F357 is clearly defined and supports the distinct side-chain orientations observed between the pH 3 and pH 4 structures. Furthermore, the Q-scores, which quantify map–model agreement, are 0.65 for pH 3 and 0.67 for pH 4, consistent with reliable assignment of the F357 orientations.

Line 258: “ K131 mutations had no discernable effect on E148A Cl⁻ transport rates (Fig. 5g, Table 2). These results align with the “stalled transport cycle” model for K131 mutants: when these low Cl⁻-binding mutant transporters are untethered from obligate Cl⁻/H⁺ coupled transport, Cl⁻ movement becomes faster, unimpeded by the need to wait for a stalled proton-transport step.” Can the authors develop this hypothesis? The point made in this section is unclear to me.

We have revised this paragraph to make the points clearer.

Lines 284-293: If K131 mutations slow transport as a direct result of weakened Cl⁻ binding, then they should

affect Cl⁻ transport rates in both WT and E148A backgrounds. On the other hand, if K131A mutations slow transport because they stall one of the E_{gate} steps in the coupled transport cycle (either protonation/deprotonation of E_{gate} or E_{gate}/Cl⁻ competition), then they should have no effect in the E148A background. We therefore tested CLC-ec1 K131A/E148A and CLC-ec1 K131M/E148A mutant transporters using the quantitative Cl⁻ flux assay (**Fig. 5c**), measuring only Cl⁻ since E148 transporters do not transport H⁺. K131 mutations had no discernable effect on E148A Cl⁻ transport rates (**Fig. 5g, Table 2**). These results align with the “stalled transport cycle” model: K131 mutations impair an E_{gate}-dependent step in the Cl⁻/H⁺ coupled transport cycle; when E_{gate} is inoperative (E148A), Cl⁻ transport rates are unaffected.

Fig.5: What does z score of 11.1 vs 7.9 mean, for a non-expert?

It is not straightforward to directly compare densities between two cryo-EM maps obtained from separate datasets, because the absolute intensity values can vary. The z-score provides a way to normalize map densities by expressing the signal in terms of how many standard deviations it lies above the average density of the entire map. This allows for a fairer comparison between maps. In our case, the density corresponding to the Cl⁻ ion is 11.1 standard deviations above the mean in the WT map, compared to 7.9 standard deviations above the mean in the K131A map. We added this information to the Methods section

Lines 592-599: “To compare Cl⁻ densities between WT and K131A mutant structures, we calculated z-scores. The Z-score normalizes the map density by expressing it as the number of standard deviations it lies above the average density of the entire cryo-EM map. This calculation is:

$$Z = \frac{\text{Density Value} - \text{Mean Density Value}}{\text{Standard Deviation of Density Value}}$$

By performing this normalization, the z-score as a normalized contour level provides a statistically robust measure of how confidently the Cl⁻ ion signal (e.g., 11.1) can be distinguished from the background noise, enabling a fair comparison between the WT and mutant maps when their absolute intensity scales are different.”

Line 236: “Importantly, we structured this flux-assay experiment to ensure that any H⁺ movement into the vesicles was a result of Cl⁻-dependent H⁺ pumping rather than leakage”. How did you do that? Can you be more specific? Edit: I found the explanation in the figure legend, but it would be nice to have a clearer pointer to it in the main text.

Thank you for requesting this clarification. We added the explanation to the main text.

Lines 266-270: Importantly, we structured this flux-assay experiment to ensure that any H⁺ movement into the vesicles was a result of Cl⁻-dependent H⁺ pumping rather than leakage: by imposing a 2-fold gradient for H⁺ on the vesicles, any leak will involve movement of H⁺ out of the vesicles, and any H⁺ movement into the vesicles must occur via CLC-ec1 actively transporting H⁺ (**Fig. 5c**).

Response to remaining Reviewer comments

We thank all the reviewers for the time and effort they devoted to re-reviewing our manuscript. Below we provide our responses to Reviewer 1's two remaining points.

- 1) This manuscript combines experimental and regular MD simulations to investigate the behavior of CIC-ec1 at pH 3. Based on the presented data, it appears that CIC-ec1 may operate under a distinct mechanism at this very acidic pH, compared to its behavior in the physiologically relevant pH range 4.5–7. The authors propose an interesting model for this low pH behavior. However, many mechanistic details remain unverified – especially concerning the proton transport, which is largely speculative at this stage. For instance, the potential involvement of E113, E202, and E203 in proton transfer is not discussed. The intriguing possibility that Cl⁻ and H⁺ may share the same conduction pathway (i.e., the pathway 1 from MD simulations), to my knowledge, has not been previously addressed in the literature. This gap in treatment is likely be due to the authors' limited expertise in reactive simulations (that include explicit proton shuttling). That being said, this paper, along with Alessio Accardi's 2024 NSMB work, suggests that CIC-ec1 may behave differently under more acidic pH values, and thus presents a useful starting point for future studies by the community. However, further reactive MD computational studies will be important in validating the proposed low-pH mechanism and this should be stated.

We regret that we were not sufficiently thorough in acknowledging the additional MD studies that will be important. That reactive MD computational studies will be important in validating our proposed mechanism has now been explicitly incorporated into our revised discussion (addition highlighted):

“Our model posits that proton transport occurs along connected water wires identified in our MD simulations. These contiguous hydrogen-bonded chains provide the structural and electrostatic continuity required for Grotthuss-like hopping through a hydrophobic protein, marking the minimal requirement for proton transport. In our simulations, we did not model an excess proton in the water wires, and we did not simulate explicit proton transport; these simplifications allowed us to focus computational effort on sampling structural dynamics and hydration patterns. In simulations of the CLC-ec1 high-pH conformation, inclusion of an excess proton was found to draw additional solvated water into the protein core and create broader water-wire structures compared to simulations without the excess proton⁵¹. Explicit proton-inclusive simulations^{40, 51, 52} would be an appropriate next step to quantify proton hopping kinetics and free-energy barriers, building on the structural and functional insights established here. Beyond this, reactive MD computational studies will be important for validating the proposed mechanism, as its ability to capture protonation and deprotonation events at specific residues provides mechanistic detail for assessing the dynamic interplay between proton transfer and protein conformational changes, complementing experimental studies.”

We have also considered the concern that symmetric pH 3 conditions are not physiological. We acknowledge this point but note that the defining functional hallmark of CLC-ec1 – the 2:1 transport stoichiometry – remains unchanged under these conditions. In this context, with the stoichiometry intact, a distinct mechanism at symmetric low pH need not be invoked.

- 2) However, I find the authors' response to my comment #10 to be inappropriate. They state: “Importantly, that modeling used structures of an inactive CLC-ec1 conformation and predated recognition that Egate can adopt an ‘out’ conformation; updating the modeling could clarify how these alternate pathway ensembles might complement the mechanism we propose.” While it is true that Mayes et al. built their kinetic model from free energy calculations at pH 7, the model was extrapolated to other pH conditions using experimental data, resulting in a plausible mechanism within the physiologically relevant pH range (Mayes et al., *J. Am. Chem. Soc.* 2018, 140, 1793–1804). In contrast, the “out” Egate conformation is not stable within this pH range (Yue et al., *Biophys. J.* 2023, 122, 1068–1085). The protein conformation reported in the current manuscript was obtained beyond this pH range. To the best of my knowledge, I have not seen decisive evidence ruling that the classical picture is incorrect within the physiologically relevant pH range. While updating the kinetic model to account for acidic pH is indeed a worthwhile goal, criticizing it for not encompassing non-physiological pH regimes is, in my opinion, unjustified. This should be revised.

We had attempted to convey the kinetic modeling in a positive light and had not intended to criticize it. We have changed wording to avoid this impression. We respectfully disagree with the conclusion that the E_{gate} conformation is not relevant within the physiologically relevant pH range; to be balanced, we have expanded the discussion of this point in this paragraph (revisions highlighted):

The proposed transport mechanism (**Fig. 6**) is relevant to CLC transporters across all kingdoms, which share fundamental features including 2:1 anion/proton coupling and E_{gate} as an essential coupling element³. The mechanism elucidates how transport occurs in the direction of Cl^- ions moving inward across the membrane and H^+ ions moving outward, which is the direction of physiological relevance for CLC-ec1 in facilitating extreme acid tolerance⁴ and for mammalian CLCs in acidifying intracellular compartments^{8, 48}. An alternative to our proposed 2:1 mechanism was previously suggested through elegant kinetic modeling, showing that multiple distinct pathways can collectively yield the experimentally observed stoichiometry⁴⁹. Because those models were built on the CLC-ec1 structures available at the time, which were recently proposed to represent an inactive conformation¹¹, extending this modeling with the additional structures now available could provide valuable insight into how ensembles of pathways may operate in CLC transporters. A point of debate in such extensions is the stability and relevance of the E_{gate} “out” conformation: while one computational study questioned its stability at physiological pH⁵⁰, other work has argued for its relevance^{10, 11, 24, 28}. Although E_{gate} “out” has been detected only under symmetric low-pH conditions, these conditions are experimentally more accessible than asymmetric pH and still preserve the hallmark 2:1 Cl^-/H^+ exchange ratio, supporting the view that the fundamental coupling mechanism remains intact and, by extension, that the findings are relevant under physiological asymmetric pH. Looking ahead, integration of complementary structural, computational, and kinetic modeling approaches will be valuable for shaping our picture of the 2:1 coupling mechanism, building on the insights advanced here.

This manuscript explores the long-debated question of whether the Cl⁻/H⁺ antiporter from *E. coli* (CIC-ec1) undergoes large-scale conformational changes or operates via stepwise ion movement within the protein. While CIC-ec1 has been extensively characterized at pH 4 and above, a few studies have suggested acid-induced conformational shifts. This study combines wet-lab experiments with molecular dynamics (MD) simulations to investigate CIC-ec1 behavior at lower pH, particularly pH 3. The authors present new cryo-EM structures of wild-type CIC-ec1 and analyze them alongside previously reported structures. Their data reveal a pH-dependent shift in the sidechain rotamers of the essential residue E148 (Glu_{ex} or E_{gate}), shedding light on its role in Cl⁻/H⁺ coupling. The pH 3 structure displays notable conformational changes, including inner gate opening, which may help fill gaps in the protein's functional cycle.

The work is interesting and relevant to the community. However, there is a discrepancy between what the obtained data demonstrates/supports and what is proposed in terms of mechanisms and explanations. The latter should be given appropriate context as primarily only ideas of what could happen and be more clearly delineated from mechanistic conclusions that are solidly supported by the authors findings. Credit to other work in the field is lacking as well and should be improved. Collectively, this work could be published once context, credit, and the following questions are addressed.

- 1) The physiological relevance of the pH 3.0 structure is unclear for a few reasons:
 - a. In extreme acid response, only the outside of the bacteria is at such low pH. Presumably the inside remains close to pH5 or the bacteria die. This should be discussed and the implications for the findings considered.
 - b. The HDX-MS suggests that large regions of the protein are water accessible at pH3.0, including TM regions that should not be water accessible. How is this physiologically possible? Is it possible the protein is only partially stable and the cryo structures are coming from a subset of the population? This should be discussed in the paper.
 - c. Given the significant change in HD exchange between pH 3 and 4.5, electrophysiology at pH 3.0 is needed. Is the stoichiometry still 2:1? How does current change?
- 2) A more direct comparison between a recently published pH 4.5 structure (Forte et al.) and the presented structures would be an important addition to understand what is truly new here. Aside from the partially obtained twist structure in that paper, how are the obtained structures different or similar to the full turn structure? How do the regions considered to be important differ?
- 3) The proposed uncoupling shows E_{gate} moving from the “middle” to the “up” position, allowing Cl⁻ to flow out without protonation. What data supports this notion? Previous work has demonstrated that this should be prohibitively unfavorable. So, why would the stalled proton transport state not be the protonated form of E_{gate}—especially at these low pH values?
- 4) The K131 results are interesting. It is argued that the impacts of this mutant are due to the decreased affinity of Cl⁻ to the central binding site. But the stability of E_{gate} in this site would also be altered. Since there is no change in the rates for E148A, the rate of release from the central site is not limiting in the absence of proton coupling. Yet the rate of transport when coupling is present is slowed. So, this mutant stalls a proton-coupled step and its impact on the stability of E_{gate} would be a more logical explanation.

- 5) The simulations show that once E_{gate} is deprotonated in the down_{out} conformation (i.e., deprotonation of state B in Fig 6) it rapidly moves to the central Cl⁻ site and pushes two Cl⁻ out. This is not consistent with the cycle shown in Fig 6 which instead shows multiple metastable intermediates D and E, which makes the latter hard to interpret and connect with the simulation results. What justifies these intermediates?
- 6) Given that the transition from B to D (with an additional Cl⁻ release to the outside) is almost instantaneous, this step must be downhill and nearly barrierless. What would cause the system to reverse? I.e., how would it go uphill from B to A?
- 7) If it really is reversible then simulations starting in A should also push Cl⁻ in. What happens when simulations start in A? Was B to A ever observed? I.e., does protonated E148 ever relocate to above the bound Cl⁻ such that deprotonation could push Cl⁻ in.
- 8) How would the transmembrane voltage influence the proposed steps in Fig 6? Was this factor included in the simulations?
- 9) What are the residues with a very large RMSD relative to the QQQ structure and why are they so different?
- 10) The reported stoichiometry at pH 4.5 (2.1 ± 0.1 , Table 2) aligns with the classical value of 2.2, yet the authors consistently refer to it as 2. Are the authors suggesting an exact 2:1 exchange ratio? Clarification is important, as a non-integer ratio indicates multiple co-existing exchange pathways (Not cited but should be: Mayes et al., *J. Am. Chem. Soc.* 2018, 140, 1793–1804). Or is there a detail I may have overlooked?
- 11) Figure 1 links extracellular pore conformation to E_{gate} rotamers, i.e., “open” with E_{gate} “out”, and “closed” or “occluded” with other E_{gate} rotamers. But in the proposed mechanism (Figure 6), all E_{gate} rotamers appear associated with an “open” extracellular pore. Are the authors proposing that the extracellular gate remains open throughout the cycle? This apparent inconsistency needs clarification.
- 12) The mechanism behind inner gate opening below pH 4 is not addressed. Is this linked to protonation of residues such as E203? Prior computational studies indicate protonation of E113 and E203 at pH 3 disrupts ion pairs with R28' from the twin monomer (Yue et al., *Biophys. J.* 2023, 122, 1068–1085). This should trigger gate opening and water entry. The absence of helix A in the QQQ structure may further support this. Do the authors have evidence supporting or refuting a role for E113, E202, or E203 protonation in this process? If so, where do the protons go once the gate opens?
- 13) The manuscript frequently refers to “water wires” and quantifies their presence (Figure 4e), yet I could not locate a clear definition. Please define what is meant by a “water wire” in this context. I note that it has been pointed out a number of times in the literature that a water wire without an excess proton in it is quite different from simply a water wire (see Li et al., *Proc. Natl. Aca. Sci.* 2021, 118, e2113141118). This should be discussed and cited.
- 14) MD simulations of the pH 3 structure identified three water wires (Figure 4a). The authors highlight that water wire 1 runs through the Cl⁻ permeation pathway and suggest its involvement in the counterclockwise

transport cycle (Figure 6). Under this scenario, do protons supposedly permeate through the water wire 1? That is, do Cl^- and H^+ share the same permeation pathway?

- 15) While water wires (without excess protons in them) are often used to infer or guess proton transport (PT) behavior, their presence alone does not guarantee PT. For example, aquaporins form water wires but block proton flow. Proton conduction involves the formation of transient hydrogen-bonded chains that excess protons themselves help assemble, as shown in CIC-ec1 (Lee et al., *Biophys. J.* 2016, 110, 1334–1345). I recommend the authors consult relevant literature (Peng et al., *J. Phys. Chem. B* 2015, 119, 9212–9218; Li et al., *Proc. Natl. Aca. Sci.* 2021, 118, e2113141118) and references therein. They should also point out that they are not simulating explicit proton transport with their MD.

In summary, while CIC-ec1 is generally thought to operate without major conformational changes within the typical pH range of 4–7.5, experimental evidence here suggests altered behavior below pH 4. This study may provide insights into the structure and function of CIC-ec1 under more acidic conditions. However, the PT mechanism at pH 3 remains unclear. I encourage the authors to not only address the issues listed above but to also expand their discussion on this issue and suggest future directions to address the remaining open questions.

Overall, I appreciate the authors' mostly satisfactory responses to our comments. However, two issues remain:

1. This manuscript combines experimental and regular MD simulations to investigate the behavior of ClC-ec1 at pH 3. Based on the presented data, it appears that ClC-ec1 may operate under a distinct mechanism at this very acidic pH, compared to its behavior in the physiologically relevant pH range 4.5–7. The authors propose an interesting model for this low pH behavior. However, many mechanistic details remain unverified – especially concerning the proton transport, which is largely speculative at this stage. For instance, the potential involvement of E113, E202, and E203 in proton transfer is not discussed. The intriguing possibility that Cl⁻ and H⁺ may share the same conduction pathway (i.e., the pathway 1 from MD simulations), to my knowledge, has not been previously addressed in the literature. This gap in treatment is likely due to the authors' limited expertise in reactive simulations (that include explicit proton shuttling). That being said, this paper, along with Alessio Accardi's 2024 NSMB work, suggests that ClC-ec1 may behave differently under more acidic pH values, and thus presents a useful starting point for future studies by the community. However, further reactive MD computational studies will be important in validating the proposed low-pH mechanism and this should be stated.
2. However, I find the authors' response to my comment #10 to be inappropriate. They state: *“Importantly, that modeling used structures of an inactive CLC-ec1 conformation and predated recognition that E_{gate} can adopt an ‘out’ conformation; updating the modeling could clarify how these alternate pathway ensembles might complement the mechanism we propose.”* While it is true that Mayes et al. built their kinetic model from free energy calculations at pH 7, the model was extrapolated to other pH conditions using experimental data, resulting in a plausible mechanism within the physiologically relevant pH range (Mayes et al., *J. Am. Chem. Soc.* 2018, 140, 1793–1804). In contrast, the “out” E_{gate} conformation is not stable within this pH range (Yue et al., *Biophys. J.* 2023, 122, 1068–1085). The protein conformation reported in the current manuscript was obtained beyond this pH range. To the best of my knowledge, I have not seen decisive evidence ruling that the classical picture is incorrect within the physiologically relevant pH range. While updating the kinetic model to account for acidic pH is indeed a worthwhile goal, criticizing it for not encompassing non-physiological pH regimes is, in my opinion, unjustified. This should be revised.